# SPARSE MOE AS A NEW RETRIEVER: ADDRESSING MISSING MODALITY PROBLEM IN INCOMPLETE MULTIMODAL DATA

## ABSTRACT

In multimodal machine learning, effectively addressing the missing modality scenario is crucial for improving performance in downstream tasks such as in medical contexts where data may be incomplete. Although some attempts have been made to effectively retrieve embeddings for missing modalities, two main bottlenecks remain: the consideration of both intra- and inter-modal context, and the cost of embedding selection, where embeddings often lack modality-specific knowledge.. In response, we propose `MoE-Retriever`[1], a novel framework inspired by the design principles of Sparse Mixture of Experts (SMoE). First, `MoE-Retriever` samples the relevant data from modality combinations, using a so-called supporting group to construct intra-modal inputs while incorporating inter-modal inputs. These inputs are then processed by Multi-Head Attention, after which the SMoE Router automatically selects the most relevant expert, i.e., the embedding candidate to be retrieved. Comprehensive experiments on both medical and general multimodal datasets demonstrate the robustness and generalizability of `MoE-Retriever`, marking a significant step forward in embedding retrieval methods for incomplete multimodal data.

## 1 INTRODUCTION

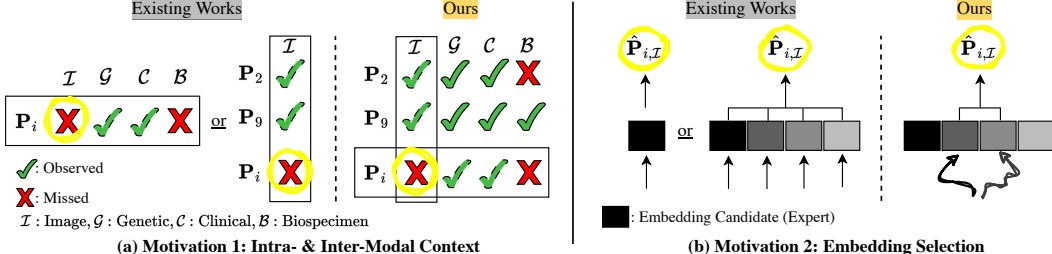

Figure 1: Motivation of this work. (a) **Motivation 1: Intra- & Inter-Modal Context**: Existing works typically consider either the intra-modal context (between samples with the same missing modality, such as $\mathbf{P}_{2,\mathcal{I}}, \mathbf{P}_{9,\mathcal{I}}$) or the inter-modal context (within a sample's observed modalities, such as $\mathbf{P}_{i,\mathcal{G}}$, $\mathbf{P}_{i,\mathcal{C}}$). In contrast, our work considers both contexts simultaneously to effectively retrieve the most relevant embedding. (b) **Motivation 2: Embedding Selection**: When retrieving the most relevant embedding ($\hat{\mathbf{P}}_{i,\mathcal{I}}$), existing approaches either use a single static embedding or combine multiple embeddings with simple methods (e.g., sum, average, attention), which makes it difficult to obtain specialized knowledge and requires activation of each embedding candidate every time. In contrast, our work leverages the design principles of SMoE, using a router to automatically select the most relevant experts through top-k selection in a sparse and efficient manner.

In the realm of multimodal machine learning, effectively handling the *missing modality scenario* has become a pivotal challenge for enhancing downstream task performance (Baltrušaitis et al., 2018; Guo et al., 2019; Wu et al., 2024a). In practical cases such as clinical and biological settings, modalities

---

[1]Source code can be found in the Supplementary Material.

such as imaging, genetic, and clinical data often contain missing entries due to varying acquisition times, costs, or patient-specific factors (Ma et al., 2021; Zhang et al., 2022a;b; Wang et al., 2023). To address this, prior approaches primarily focus on two strategies: imputing missing features directly within the input feature space or employing learnable embedding to represent missing features in the latent space. The former often involves some rule-based prior, such as using the population mean to perform imputation. This method does not scale with data, as the imputation method remain fixed when the underlying distribution changes. In contrast, recent research has increasingly turned toward the latter — leveraging learnable embedding to provide more adaptive and context-aware representations for missing modalities (Zhang et al., 2022b; Wang et al., 2023; Zhang et al., 2022a; Wu et al., 2024b; Han et al., 2024). However, despite their promise, these learnable embedding-based methods still face several critical limitations.

**Intra- & Inter-Modal Context.** As illustrated in Figure 1 (a), current methods inadequately address both intra-modal and inter-modal contexts when supplementing missing modalities, often focusing on one or the other. In intra-modal scenarios, the goal is to retrieve embeddings for the missing (target) modality by identifying similar samples (Malitesta et al., 2024). However, existing works often choose unimodal approaches that primarily address intra-modal context, failing to personalize the sample's heterogeneous context. Conversely, in inter-modal scenarios, it is assumed that modality-invariant and modality-specific information exists across input modalities, suggesting that missing modalities can be imputed from the sample's specific observed modalities (Zhang et al., 2022b; Wang et al., 2023). However, these works do not carefully consider intra-sample information while proceeding with multi-modal fusion. As a result, focusing solely on either intra-modal or inter-modal context leads to incomplete supplementation and limits the model's ability to effectively leverage the rich multimodal information available in real-world datasets. This highlights the need for a more holistic approach that integrates both perspectives for more accurate and robust imputation of missing modalities.

**Embedding Selection.** Figure 1 (b) illustrates the current state of embedding retrieval. Current methods either treat the learnable or retrieved embeddings as a single embedding (Wang et al., 2023; Han et al., 2024) or use diverse embeddings but require activating all candidates every time a retrieval is performed, using operations like summation, averaging, or attention mechanisms. These methods can incur a high computational cost as the number of samples or modalities grows, and they lack the ability to adapt to diverse observed modality combinations, treating all potential scenarios equally regardless of the specific context. This uniformity in handling observed modalities limits the capacity for more nuanced and context-specific supplementation. For instance, specific knowledge may be required when certain input modality combinations are present, which is crucial for improving downstream task performance.

**Our Approach.** To address these challenges, we propose `MoE-Retriever`, a novel framework for embedding retrieval given a incomplete multimodal data. The main idea of `MoE-Retriever` is to borrow the desgin principle from the Sparse Mixture of Experts (SMoE), which activates most relevant experts (i.e., embedding candidates) given a specific intra- and inter-modal context within in a router in a sparse manner. To achieve this, we first begin with genertaing supporting group which is based on given modality combination and aim for target (missing) modality, responsible for sampling intra-modal samples. Next by incoporating inter-modal samples and via Multi-head attention within this incoproated inputs and router with experts which both includes the shared and modality-specific experts finally retrieves the most relevant embedding for target modality. Extensive experiments on two medical datasets (ADNI, MIMIC) and two general machine learning datasets (ENRICO, CMU-MOSI) validate the efficacy and generalizability of our proposed method, consistently demonstrating its robust performance across various multimodal settings.

- We highlight that current intra- or inter- modal or single or multiple-but-lacking specialized knwoledge brings the bottleneck into incomplete multimodal embedding retrieval.

- We propose `MoE-Retriever`, borrowing the design principle of Sparse Mixture of Experts design, which inputs the both intra-modal inter-sample and inter-modal intra-sample contexts and retrieve most relevant embedding from modality-specific and shared experts.

- Our comprehensive experimental evaluations on the medicinal dataset and machine learning datasets, showcase the effectiveness and portability of `MoE-Retriever`.

## 2 RELATED WORK

**Multimodal Learning with Missing Modality.** Multimodal learning has garnered increasing attention in the machine learning community, particularly in the medical domain, where clinical data is inherently multimodal (Khader et al., 2023; Steyaert et al., 2023). However, in real-world clinical practice, missing modalities are a common challenge (Zhou et al., 2023; Liu et al., 2023). To address this issue, one straightforward approach is to leverage generative models to impute the missing modalities (Pan et al., 2021; Zhang et al., 2024). Nonetheless, generative modeling of another distribution is a ill-posed problem (Zhang et al., 2022a). In contrast, non-generative approaches have emerged, utilizing techniques such as graph-based modeling (Wu et al., 2024b), and modality fusion (Zhang et al., 2022b; Wang et al., 2023; Yao et al., 2024). While these methods can harness both inter-patient and intra-patient information, they face challenges related to scalability and struggle to handle fleximodal scenarios (Han et al., 2024), where any combination of modalities may be present. To improve scalability, FuseMoE (Han et al., 2024) introduced a sparse Mixture-of-Experts (MoE) model aims to be robust to any combination of missing modality scenario. However, despite its scalability advantages, FuseMoE do not explicitly account both the inter-patient and intra-patient relationships simultaneously, limiting its ability to fully utilize the multimodal context of clinical data.

**Sparse Mixture-of-Experts (SMoE).** SMoE (Shazeer et al., 2017) builds on the traditional Mixture-of-Experts (MoE) model (Jacobs et al., 1991; Jordan & Jacobs, 1994; Chen et al., 1999; Yuksel et al., 2012) by introducing sparsity, which enhances both computational efficiency and model performance. By selectively activating only the most relevant experts for a specific task, SMoE minimizes overhead and improves scalability, making it particularly useful for complex, high-dimensional datasets across various applications. It has been widely applied in both vision (Riquelme et al., 2021; Lou et al., 2021; Ahmed et al., 2016; Wang et al., 2020; Yang et al., 2019; Abbas & Andreopoulos, 2020) and language processing (Lepikhin et al., 2021; Kim et al., 2021; Zhou et al., 2022; Zhang et al., 2021; Zuo et al., 2022; Jiang et al., 2021). Its capacity to dynamically allocate different network parts to specific tasks (Ma et al., 2018; Aoki et al., 2021; Hazimeh et al., 2021; Chen et al., 2023) or data modalities (Kudugunta et al., 2021) has been explored for various applications (Mustafa et al., 2022). Research shows its effectiveness in areas like classification tasks for digital number recognition (Hazimeh et al., 2021) and medical signal processing (Aoki et al., 2021). However, the current use of SMoE is often biased toward its role as a backbone design, typically integrated into Transformer architectures to improve embedding representations in fusion or prediction layers. Its potential for more effective use, such as serving as a retriever or supplementing missing embeddings to bridge the feature space and encoder space, remains underexplored.

## 3 METHOD

### 3.1 PRELIMINARIES AND NOTATIONS

**Motivation behind bringing SMoE design.** In the context of incomplete multimodal data, only the observed features in the raw feature space can pass through the modality-specific encoder. This raises a critical question: *how can we effectively handle samples with missing modalities to provide robust embeddings for the missing features?* Ensuring that the embedding space, followed by the fusion and prediction layers, remains trainable through continuous gradient flow is essential. It is important to note that different samples exhibit varying combinations of observed modalities, which necessitates a personalized approach capable of handling each sample's unique environment, such as its specific modality combination.

To address this challenge, we introduce the design principles of SMoE (Shazeer et al., 2017). Given a pool of diverse experts (i.e., trainable feed-forward networks), the SMoE architecture enables the automatic and sparse activation of different experts, each specializing in certain knowledge, based on the input scenario. This dynamic routing mechanism effectively mitigates the limitations of static, one-size-fits-all designs, where learnable embeddings are constrained to a single expert or a fixed combination of embeddings without a router. In such static setups, embeddings for missing modalities are often selected at random, leading to suboptimal performance for downstream tasks.

**Notation.** Formally, SMoE consists of multiple experts, denoted as $\mathcal{E}_1, \ldots, \mathcal{E}_{|\mathcal{E}|}$, where $|\mathcal{E}|$ represents the total number of experts, and a router, $\mathcal{R}$, which governs the routing mechanism, sparsely selecting

the top-$k$ experts. For a given embedding or token $\mathbf{x}$, the router $\mathcal{R}$ activates the top-$k$ experts based on the highest scores derived from a softmax function applied to the outputs of a learnable gating function, $g(\cdot)$, typically modeled as a one or two-layer MLP. The router's output, $\mathcal{R}(\mathbf{x})_i$, indicates the selection of the $i$-th expert. This process is formally described as follows:

$$
\begin{aligned}
\mathbf{y} &= \sum_{i=1}^{|E|} \mathcal{R}(\mathbf{x})_i \cdot \mathcal{E}_i(\mathbf{x}), \\
\mathcal{R}(\mathbf{x}) &= \text{Top-K}(\text{softmax}(g(\mathbf{x})), k), \\
\text{TopK}(\mathbf{v}, k) &= \begin{cases} \mathbf{v}, & \text{if } \mathbf{v} \text{ is in the top } k, \\ 0, & \text{otherwise.} \end{cases}
\end{aligned}
\tag{1}
$$

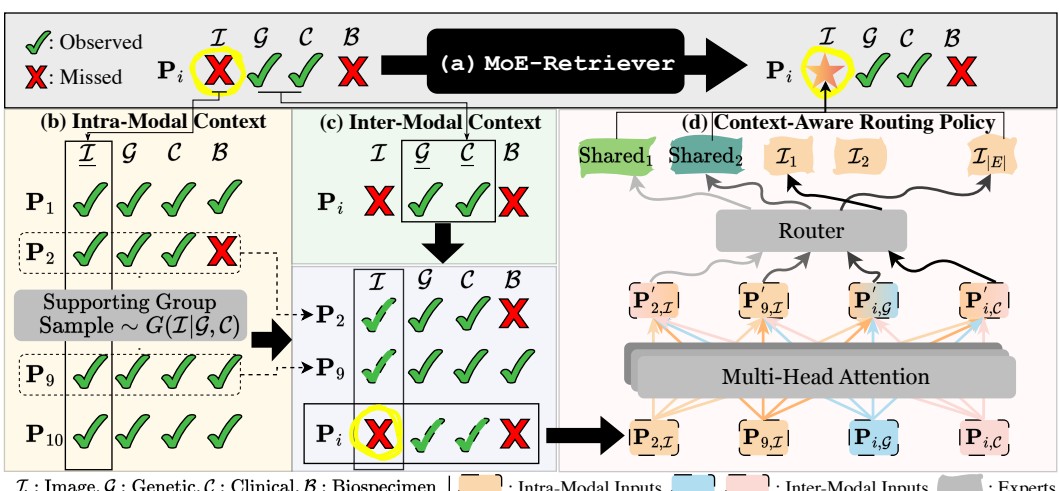

Figure 2: Overall illustration of `MoE-Retriever`. (a) **The role of `MoE-Retriever`.** Given a sample ($\mathbf{P}_i$) with a missing modality, $\mathcal{I}$ (Image), our goal is to retrieve the most relevant embedding ($\mathbf{P}_{i,\mathcal{I}}$) by considering two contextual factors. First, we focus on (b) **Intra-Modal Context**, which seeks to find embeddings within the same modality as the missing one ($\mathcal{I}$) to reflect similar contextual knowledge. To achieve this, we define a supporting group ($G(\mathcal{I}|\mathcal{G},\mathcal{C})$), where the target modality ($\mathcal{I}$) and the sample's observed modalities ($\mathcal{G}, \mathcal{C}$) form a sufficient context for grouping. After sampling from this group, we incorporate the sample's specific (c) **Inter-Modal Context**, leveraging the observed modalities. We then proceed to (d) **Context-Aware Routing Policy**, which first applies multi-head attention and adopts the SMoE framework. Here, the router (top-1 selection in this example) selects the most relevant experts given two intra- and inter-modal inputs. After integrating all the embeddings, the final embedding is regarded as the retrieved embedding for the sample $i$'s missing modality $\mathcal{I}$, denoted as $\mathbf{P}_{i,\mathcal{I}}$. For retrieving an embedding for another missing modality, $\mathcal{B}$, the supporting group would be updated to $G(\mathcal{B}|\mathcal{G},\mathcal{C})$, and the intra-modal embeddings would consist of $\mathbf{P}_{\cdot,\mathcal{B}}$, with the expert selection adapted accordingly to $\{\mathcal{B}_1, \ldots, \mathcal{B}_E\}$.

## 3.2 MoE-Retriever

The overall framework of `MoE-Retriever`, along with the detailed procedure, is illustrated in Figure 2. In essence, the key idea behind `MoE-Retriever` is to retrieve the most relevant embedding for the missing modality by leveraging two contexts: (1) **Intra-Modal Context**, which samples similar examples from a well-defined supporting group based on the observed modality combination (Sec 3.2.1), and (2) **Inter-Modal Context**, which considers the sample-specific heterogeneous combination of observed modalities (Sec 3.2.2). The next step is (3) **Context-Aware Routing**, where the expert pool is designed modality-specifically, using both contexts to effectively supplement the target (i.e., missing) modality. Finally, the selected experts and their linear combination with the inputs are integrated into a single embedding, which is regarded as the final retrieved embedding (Sec 3.2.3).

### 3.2.1 Intra-Modal Context

We begin with the *intra-modal context* (column-wise context in Figure 2), where intra-modal refers to the homogeneous modality that matches the target modality we aim to supplement. The rationale for incorporating this context is that, by forming a pool of similar samples, we can capture patterns directly observed across patients, without requiring any additional preprocessing. The observed pattern can be represented as a *modality combination*, which reflects similar trends or patterns, i.e., knowledge observed across the samples. Empirically, samples (e.g., patients) with similar observed modality combinations have shown exhibit analogous characteristics. For instance, patients who lack the image modality but possess both genetic and clinical modalities may be more likely to display correlations with certain domain-specific traits, such as early-stage diagnosis, mild cognitive impairment, or slower progression rates, often associated with genetic risk factors like the APOE $\epsilon 4$ allele (Dubois et al., 2023; Jack Jr et al., 2018; Lambert et al., 2013).

To effectively sample from an intra-modal sample pool, we first need to define a modality combination-specific pool, which we denote as the *supporting group*. The core idea behind the supporting group is that, given an observed modality combination and a target (missing) modality, the corresponding group must include the observed modalities as well as the target modality to support the patient's intra-modal pool. For example, if a sample contains the modalities '$\mathcal{GC}$' and we aim to impute the modality '$\mathcal{I}$' (as illustrated in Figure 2), the supporting group should include samples with 'GC' as well as the missing modality '$\mathcal{I}$'. Consequently, the supporting group would comprise samples with modality combinations such as '$\mathcal{IGC}$' or '$\mathcal{IGCB}$'.

Formally, as an example from Figure 2, let the set of modalities be $\mathcal{M} = \{\mathcal{I}, \mathcal{G}, \mathcal{C}, \mathcal{B}\}$. With a specific modality combination $mc \in \mathcal{MC} = \{\mathcal{I}, (\mathcal{I}, \mathcal{G}), (\mathcal{I}, \mathcal{G}, \mathcal{C}), \ldots, \mathcal{G}, (\mathcal{G}, \mathcal{C}), \ldots, (\mathcal{I}, \mathcal{G}, \mathcal{C}, \mathcal{B})\}$, where the total number of combinations in $\mathcal{MC}$ is $|\mathcal{MC}| = \sum_{m=1}^{|\mathcal{M}|-1} \binom{|\mathcal{M}|}{m} = 2^{|\mathcal{M}|} - 1$, the supporting group $G$ consists of the samples that satisfy the following constraints:

$$G(j \mid \mathcal{T}, mc) = \{j \in \{1, 2, \ldots, N\} \mid mc_j \in \mathcal{X}(\mathcal{S} \mid \mathcal{T}, mc)\}$$
$$\text{where } \mathcal{X}(\mathcal{S} \mid \mathcal{T}, mc) = \{S \subseteq \mathcal{M} \mid (mc \subseteq S) \wedge (\mathcal{T} \in S)\} \quad \forall \mathcal{T} \in \mathcal{M}, \quad \forall mc \in \mathcal{MC} \tag{2}$$

where $G(j \mid \mathcal{T}, mc)$ denotes the set of sample indices among total sample size $N$, derived from the set of possible modality combinations $\mathcal{X}(\mathcal{S} \mid \mathcal{T}, mc)$ for a given target modality $\mathcal{T}$ and modality combination $mc$. In this context, the satisfying $\mathcal{S}$ denotes any arbitrary set of modality combinations that satisfies the constraint of including both $mc$ (i.e., $(mc \subseteq S)$) and (i.e., $\wedge$) the target modality $\mathcal{T}$ as subsets (i.e., $(\mathcal{T} \in S)$). Given the supporting group $G$, we sample[2] intra-modal examples that assist in the final retrieval by SMoE by referring to similar examples within the homogeneous modality.

### 3.2.2 Inter-Modal Context

Beyond intra-modal context, we now consider another critical dimension: inter-modal context (illustrated row-wise in Figure 2). This approach allows us to incorporate personalized context specific to a given sample that would be missed by only considering intra-modal context. As a real-world example, this perspective is particularly meaningful in multimodal medical scenarios such as Alzheimer's diagnosis. When genetic (G) and clinical (C) data are available but imaging (I) is missing (case of Figure 2), it may suggest the patient is in the early stages of the disease, where less invasive and more accessible modalities are prioritized. Imaging, typically more expensive, may be reserved for later stages when symptoms progress (Dubois et al., 2023; Li et al., 2022). Additionally, genetic and clinical data alone can provide valuable early insights, guiding initial interventions before resorting to costly imaging techniques (Kim, 2023).

Formally, to consider inter-modal context, we directly focus on the observed modalities, i.e., $mc$ (e.g., $(\mathcal{G}, \mathcal{C})$) for a sample index, $i$. By doing so, we integrate these sample-specific heterogeneous modality combinations, which will serve as input for the inter-modal examples in the final retrieval by SMoE, referring to the personalized context within the heterogeneous modalities.

---

[2]For the number of samples, we used a count that matches the observed modalities of the samples (i.e., $|mc|$) to ensure a balanced impact of both. They may vary and can be treated as a hyperparameter for flexibility. However, empirical observations indicate that varying the number of intra-modal samples has only a marginal effect on model performance.

### 3.2.3 CONTEXT-AWARE ROUTING POLICY

Now, given two contexts, i.e., intra-modal and inter-modal, we proceed with context-aware routing via the SMoE design. The goal of this routing is to retrieve the most relevant expert given an input combination that includes both homogeneous and heterogeneous modality information. For each embedding (i.e., token) input to the router, the router is trained to select the most relevant expert that can benefit the downstream task. The selected experts are expected to specialize in handling the specific input modalities.

The context-aware router design is detailed as follows:

$$\hat{\mathbf{P}}_{i,\mathcal{T}} = \sum_{e=1}^{|E|} \mathcal{R}(\mathbf{x})_e \cdot \mathcal{E}_e^{\mathcal{T}}(\mathbf{x}) \tag{3}$$

$$\text{where} \quad \mathbf{x} \in \{\mathbf{P}'_{i_{\text{intra}},\mathcal{T}} \cup \mathbf{P}'_{i,mc}\}, \quad \forall i_{\text{intra}} \in G(\mathcal{T} \mid mc), \forall \mathcal{T} \in \mathcal{M}, \quad \forall mc \in \mathcal{MC}$$

where $\hat{\mathbf{P}}_{i,\mathcal{T}}$ is the predicted retrieved embedding for sample $i$'s missing modality $\mathcal{T}$. $\mathcal{R}(\cdot)$ denotes the router responsible for top-k expert selection, as defined in Equation 1, given an input embedding or token. Here, the input of SMoE, $\mathbf{x}$ includes (i.e., $\cup$) both intra-modal examples ($\mathbf{P}'_{i_{\text{intra}},\mathcal{T}}$) and inter-modal examples ($\mathbf{P}'_{i,mc}$). $\mathbf{P}' = \text{MHA}(\mathbf{P})$, where $\mathbf{P}$ represents the embedding after passing through the modality-specific encoder from raw feature space. This denotes the embedding or token after undergoing Multi-Head Attention (MHA), i.e., Cross-Attention, enabling interaction between tokens. Thus, tokens are endowed with not only self-modality knowledge but also inter-modal harmonization before being passed to the SMoE router.

For the expert design, $\mathcal{E}_e^{\mathcal{T}}(\mathbf{x})$ represents the modality-specific expert , where each expert corresponds to a distinct FFN layer, is distinct and newly introduced in `MoE-Retriever` to enhance context-awareness, particularly in handling missing modality scenarios. Notably, the retrieval target differs for each modality combination in various samples, leading us to allocate specific expert indices for each target modality. For instance, if there are 32 experts and four modalities, each modality will have its own pool of 8 experts. Additionally, to enhance flexibility and generalizability, we include shared experts (denoted as 'Shared' in Figure 2), expecting that common knowledge can be leveraged across different modalities. The number of shared experts is controlled by the hyperparameter $b$, and we elaborate on this design in Section 4.4.

After retrieving the most relevant embedding for each missing modality, we proceed to the subsequent fusion layer[3], followed by the prediction head for the downstream task. Since gradients flow continuously from the input features to the output predictions, this enables end-to-end training.

### 3.3 OVERALL ALGORITHM

To summarize, the overall algorithm of `MoE-Retriever` is detailed in Algorithm 1.

---

[3]The fusion layer can be based on diverse architectures, such as Transformers or even an SMoE layer. To ensure generalizability, we choose a vanilla Transformer encoder as our fusion layer and explore alternative backbones in the Experiments section.

---

**Algorithm 1** The overall procedure of `MoE-Retriever`.

---
1: **Input:** Samples, $i \leq N$, Supporting Group, $G(\mathcal{T} \mid mc)$, Modality Set, $\mathcal{M}$, Modality Combination Set, $mc$
2: **Output:** Retrieved Embedding for Missing Modality, $\mathcal{T}$
3: **for** $i = 1, \cdots, N$ **do**
4:     **if** $|mc_i| < |\mathcal{M}|$ :
5:       **for** $t \in \mathcal{T}_i$ **do**
6:         $\mathbf{x} = []$
7:         /* Intra-Modal Context */
8:         Samples $\sim G(t \mid mc_i)$
9:         **for** $j \in$ Samples **do**
10:           $\mathbf{x}$.append($\mathbf{P}_{j,\mathcal{T}}$)
11:         **end for**
12:         /* Inter-Modal Context */
13:         **for** $mc \in mc_i$ **do**
14:           $\mathbf{x}$.append($\mathbf{P}_{i,mc}$)
15:         **end for**
16:         /* Context-Aware Routing Policy */
17:         $\mathbf{x} \leftarrow \text{MHA}(\mathbf{x})$
18:         $\hat{\mathbf{P}}_{i,\mathcal{T}} \leftarrow \text{SMoE}(\mathbf{x}, \mathcal{R}, \mathcal{E}^{\mathcal{T}}, \text{top-}k)$
19:       **end for**
20: **end for**

---

## 4 EXPERIMENTS

### 4.1 EXPERIMENTAL SETTINGS

**Multimodal Medical Datasets.** We evaluate `MoE-Retriever` on two real-world multimodal medical datasets. *ADNI Dataset*: The Alzheimer's Disease Neuroimaging Initiative (ADNI) is pivotal for Alzheimer's Disease (AD) research, aggregating multimodal data on disease evolution and biomarkers including four modalities, image modality (MRI and PET scans), genetic profiles, clinical metrics, and biospecimen samples, with open access for research standardization (Weiner et al., 2010; 2017). After preprocessing, we extract 2380 samples and target a three-tier classification task of AD stages: Dementia, Cognitively Normal (CN), or Mild Cognitive Impairment (MCI). *MIMIC Dataset*: MIMIC-IV (Medical Information Mart for Intensive Care IV) is sourced from critical care units, offers both structured (demographics, vitals, labs, medications) and unstructured data (clinical notes). For the experiments, we extracted labs results, clinical notes, and ICD 9 codes from 9,003 patient records to predict a binary classification of one-year mortality prediction (Johnson et al., 2023). For the detailed preprocessing for ADNI and MIMIC dataset, please refer to Appendix A.1.

**Additional Multimodal Datasets.** To demonstrate the generalizbility of `MoE-Retriever` toward other real-world multimodal domain, we use two general multimodal datasets. *CMU-MOSI Dataset*: The Multimodal Corpus of Sentiment Intensity (CMU-MOSI) dataset comprising 2,199 annotated video clips, advances affect recognition through detailed sentiment analysis on a scale from -3 to +3, utilizing YouTube vlogs for real-world sentiment expression research (Zadeh et al., 2016). *ENRICO Dataset*: The Enhanced Rico (ENRICO) (Leiva et al., 2020) dataset is a collection of 1,460 Android app screens, each comprising an image along with the set of apps and their respective locations. This dataset is organized into 20 distinct design categories, which focuses on a classification tasks to identify different design motifs.

**Baselines.** We compare `MoE-Retriever` against various state-of-the-art baselines from three categories. (1) **feature modeling methods**: mmFormer (Zhang et al., 2022b) and ShaSpec (Wang et al., 2023)). (2) **graph-based approaches**: MUSE (Wu et al., 2024b) and M3Care (Zhang et al., 2022a). (3) **MoE-based method**: FuseMoE (Han et al., 2024). For details regarding modality-specific encoders setting, please refer to Appendix A.2.

**Implementations.** To ensure a fair comparison with other baselines, we utilized the optimal hyper-parameter settings provided in the original papers. For dataset split, we choose 70% for training, 15% as validation set, and the remaining 15% for testing. Both the ADNI and MIMIC datasets contain missing data. For the CMU-MOSI and ENRICO datasets, we applied random dropping with probability of 0.3 for each modality independently to simulate missing modality scenarios. Given the

incomplete nature of the datasets, if a baseline implementation could impute or interact with other modalities, we leveraged those methods. Otherwise, we used zero-padding to support batch-wise training. All experiments were conducted on NVIDIA A100 GPUs. Each experiment was run three times with different seeds to ensure reproducibility, and the results were averaged.

## 4.2 Primary Results

| Dataset | Modality | Metric | mmFormer | ShaSpec | M3Care | MUSE | FuseMoE | MoE-Retriever |
|---------|----------|--------|----------|---------|--------|--------|---------|---------------|
| ADNI | $\mathcal{I}+\mathcal{G}$ | Acc. | $50.42_{\pm4.98}$ | $54.81_{\pm4.47}$ | $48.69_{\pm4.03}$ | $43.90_{\pm2.59}$ | $60.41_{\pm0.87}$ | $\mathbf{61.09}_{\pm2.12}$ |
| | | F1 | $46.66_{\pm2.40}$ | $54.43_{\pm4.11}$ | $40.29_{\pm6.49}$ | $26.83_{\pm2.68}$ | $61.04_{\pm0.95}$ | $\mathbf{62.10}_{\pm1.12}$ |
| | $\mathcal{I}+\mathcal{G}+\mathcal{C}$ | Acc. | $51.73_{\pm1.40}$ | $58.36_{\pm1.65}$ | $48.97_{\pm2.45}$ | $45.04_{\pm2.65}$ | $60.97_{\pm1.32}$ | $\mathbf{63.12}_{\pm1.19}$ |
| | | F1 | $49.97_{\pm1.89}$ | $52.69_{\pm4.99}$ | $43.55_{\pm6.24}$ | $37.21_{\pm2.61}$ | $61.30_{\pm1.07}$ | $\mathbf{62.17}_{\pm2.90}$ |
| | $\mathcal{I}+\mathcal{G}+\mathcal{C}+\mathcal{B}$ | Acc. | $55.46_{\pm1.05}$ | $59.94_{\pm2.25}$ | $54.68_{\pm0.70}$ | $52.24_{\pm2.61}$ | $59.52_{\pm1.00}$ | $\mathbf{64.52}_{\pm2.55}$ |
| | | F1 | $46.94_{\pm0.31}$ | $59.94_{\pm1.88}$ | $46.09_{\pm2.29}$ | $43.07_{\pm2.01}$ | $59.55_{\pm1.60}$ | $\mathbf{63.80}_{\pm2.96}$ |
| MIMIC | $\mathcal{L}+\mathcal{N}$ | Acc. | $77.37_{\pm0.00}$ | $77.37_{\pm0.15}$ | $76.14_{\pm0.46}$ | $\mathbf{77.40}_{\pm1.12}$ | $60.50_{\pm3.82}$ | $76.82_{\pm3.02}$ |
| | | F1 | $43.62_{\pm0.00}$ | $55.19_{\pm1.52}$ | $45.26_{\pm0.44}$ | $51.53_{\pm1.90}$ | $52.79_{\pm1.32}$ | $\mathbf{58.06}_{\pm2.19}$ |
| | $\mathcal{L}+\mathcal{C}$ | Acc. | $77.37_{\pm0.00}$ | $77.37_{\pm0.13}$ | $76.76_{\pm0.59}$ | $\mathbf{77.40}_{\pm1.12}$ | $63.31_{\pm3.21}$ | $77.20_{\pm0.47}$ |
| | | F1 | $43.62_{\pm0.00}$ | $57.32_{\pm0.52}$ | $43.92_{\pm0.52}$ | $51.53_{\pm1.90}$ | $54.78_{\pm0.91}$ | $\mathbf{57.73}_{\pm0.64}$ |
| | $\mathcal{N}+\mathcal{C}$ | Acc. | $77.37_{\pm0.00}$ | $77.40_{\pm0.03}$ | $77.26_{\pm0.35}$ | $77.32_{\pm1.13}$ | $64.77_{\pm0.36}$ | $\mathbf{77.45}_{\pm0.14}$ |
| | | F1 | $43.62_{\pm0.00}$ | $54.59_{\pm0.65}$ | $45.31_{\pm1.22}$ | $51.53_{\pm1.90}$ | $55.54_{\pm0.60}$ | $\mathbf{56.65}_{\pm1.23}$ |
| | $\mathcal{L}+\mathcal{N}+\mathcal{C}$ | Acc. | $77.37_{\pm0.00}$ | $\mathbf{77.40}_{\pm0.09}$ | $76.04_{\pm0.70}$ | $77.40_{\pm1.12}$ | $63.90_{\pm1.72}$ | $76.59_{\pm0.07}$ |
| | | F1 | $43.62_{\pm0.00}$ | $55.79_{\pm0.94}$ | $45.43_{\pm1.17}$ | $51.25_{\pm1.87}$ | $55.38_{\pm0.16}$ | $\mathbf{59.74}_{\pm0.81}$ |

Table 1: Performance comparison in ADNI and MIMIC Datasets. Image ($\mathcal{I}$), Genetic ($\mathcal{G}$), Clinical ($\mathcal{C}$), and Biospecimen ($\mathcal{B}$) modalities are used for ADNI dataset. For ADNI dataset, we use the image modality as a central reference, and sequentially added genetic, clinical, and finally all four modalities. Lab ($\mathcal{L}$), Notes ($\mathcal{N}$), and Code ($\mathcal{C}$) modalities are used in MIMIC dataset. We report Accuracy (Acc.) and F1-Macro (F1) scores.

| Dataset | Modality | mmFormer | ShaSpec | M3Care | MUSE | FuseMoE | MoE-Retriever |
|---------|----------|----------|---------|--------|--------|---------|---------------|
| ENRICO | $\mathcal{S}+\mathcal{W}$ | $36.19_{\pm0.98}$ | $21.03_{\pm0.32}$ | $19.06_{\pm5.17}$ | $36.01_{\pm2.81}$ | $36.99_{\pm6.83}$ | $\mathbf{38.24}_{\pm1.16}$ |
| CMU-MOSI | $\mathcal{V}+\mathcal{A}$ | $42.23_{\pm0.00}$ | $50.91_{\pm1.63}$ | $42.23_{\pm0.00}$ | $44.64_{\pm1.94}$ | $47.46_{\pm2.36}$ | $\mathbf{53.12}_{\pm2.26}$ |
| | $\mathcal{V}+\mathcal{T}$ | $62.20_{\pm0.90}$ | $60.01_{\pm1.44}$ | $42.12_{\pm0.14}$ | $52.54_{\pm1.92}$ | $63.77_{\pm1.62}$ | $\mathbf{65.74}_{\pm0.55}$ |
| | $\mathcal{A}+\mathcal{T}$ | $65.65_{\pm0.63}$ | $65.09_{\pm1.02}$ | $47.05_{\pm6.83}$ | $50.82_{\pm1.91}$ | $61.33_{\pm0.93}$ | $\mathbf{66.13}_{\pm0.69}$ |
| | $\mathcal{V}+\mathcal{A}+\mathcal{T}$ | $62.75_{\pm1.12}$ | $64.02_{\pm0.65}$ | $42.23_{\pm0.00}$ | $50.66_{\pm1.93}$ | $60.67_{\pm0.22}$ | $\mathbf{65.21}_{\pm2.72}$ |

Table 2: Performance comparison in ENRICO and CMU-MOSI Datasets. Screenshot ($\mathcal{S}$), and Wireframe ($\mathcal{W}$) modalities are used for ENRICO dataset. Vision ($\mathcal{V}$), Audio ($\mathcal{A}$), and Text ($\mathcal{T}$) modalities are used in CMU-MOSI dataset. We report Accuracy (Acc.) for both datasets.

**Results on ADNI and MIMIC Datasets.** Table 1 presents several insights: **1)** On the ADNI dataset, among all modality combinations, MoE-Retriever outperforms all baselines by a notable margin. **2)** Notably, as the number of available modalities increases (e.g., $\mathcal{I} + \mathcal{G} + \mathcal{C} + \mathcal{B}$), the potential of MoE-Retriever grows, providing a large margin of improvement (7.64% gain compared to the best-performing model, ShaSpec, and 8.40% gain compared to the state-of-the-art model, FuseMoE). This shows that with more modalities, there is greater room for improvement, which can be attributed to the fact that a larger number of intra- and inter-modal samples facilitate the retrieval process. **3)** The two graph-based methods, M3Care (Zhang et al., 2022a) and MUSE (Wu et al., 2024b), perform the worst on the ADNI dataset. This suggests that while graph-based approaches capture intra-modal relationships between samples, they struggle due to the lack of handling inter-modal interactions, highlighting the importance of these interactions. **4)** FuseMoE (Han et al., 2024), a mixture-of-experts (MoE)-based method, achieves the best performance on the ADNI dataset but significantly underperforms on the MIMIC dataset [4]. This can be attributed to FuseMoE's reliance on a single random embedding to impute missing modalities. **5)** On the MIMIC dataset,

---

[4] We attempted to use the authors' code but observed unstable performance. Thus, we borrowed FuseMoE's performance on these datasets from the recent Flex-MoE paper (Yun et al., 2024).

all baseline models suffer from the label imbalance problem, resulting in either Acc or F1 scores being biased. However, `MoE-Retriever` appears to be a well-balanced model, where the F1 score, being more significant than Acc in imbalanced cases, consistently outperforms all baselines. All in all, `MoE-Retriever` achieves notable performance gains on both datasets, thanks to its ability to model intra- and inter-modal contexts and its context-aware routing policy via the SMoE design, showcasing that better-retrieved embeddings for missing modalities lead to downstream performance improvements.

**Results on ENRICO and CMU-MOSI Datasets.** Table 2 shows the performance across generalized domains: design motifs for the ENRICO dataset and sentiment analysis for the CMU-MOSI dataset. We observe that **1)** `MoE-Retriever` outperforms current multimodal baselines, demonstrating its generalizability across diverse multimodal domains. Specifically, in the CMU-MOSI dataset, we observe **2)** that as the number of modalities increases, the performance of existing baselines improves, but the increase does not surpass that of `MoE-Retriever`, highlighting its effectiveness as a strong benchmark model for various domains and modality combinations.

### 4.3 How MoE-Retriever Contributes?

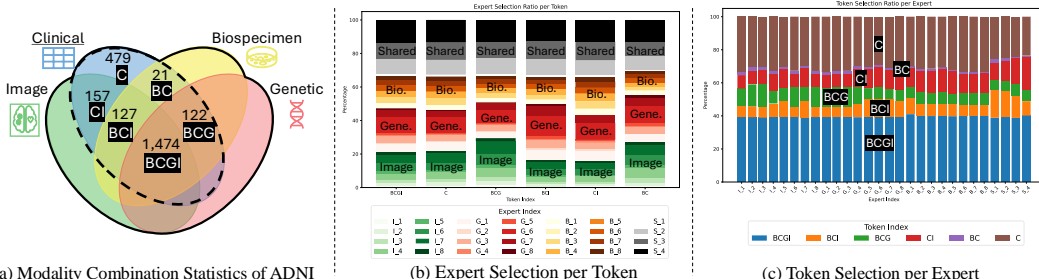

(a) Modality Combination Statistics of ADNI     (b) Expert Selection per Token     (c) Token Selection per Expert

Figure 3: (a) Statistics of modality combinations observed in the ADNI dataset. We observe that although the ADNI dataset comprises four modalities, the modality combinations are not as diverse, showing only six unique regions. Notably, all modality combinations include the clinical modality. (b) Given an input token (i.e., modality combination), we track the expert selection ratio based on the modality combination. Alternatively, (c) from the expert's perspective, we provide how each expert selects the input token and their relative ratio. The backbone illustration of (a) is adapted from (Yun et al., 2024).

**In-depth Analysis.** To gain a deeper understanding of how `MoE-Retriever` functions and contributes to embedding retrieval, we provide an in-depth analysis using the ADNI dataset in Figure 3. First, as shown in Figure 3 (a), we observe six unique modality combination regions. Interestingly, the clinical modality is present in all combinations, indicating that the input token will always include the clinical ($\mathcal{C}$) modality. This also suggests that the missing modality, i.e., the target modality, will often include $\mathcal{I}$, $\mathcal{G}$, or $\mathcal{B}$, depending on its interaction with other modalities.

Next, after training `MoE-Retriever`, we track the activation ratio from both token and expert perspectives. In Figure 3 (b), we observe: **1)** `MoE-Retriever` successfully learns which modality should be selected and imputed. For example, when the token index is given as $BCG$, which lacks the $\mathcal{I}$ modality, the majority of tokens select image-specific experts, ranging from $I_1$ to $I_8$. **2)** This imputation tendency is also observed when the input token is $BCI$ or $BC$, naturally incorporating the missing modality. This indicates that both the router and the experts are equipped with the knowledge of how to handle different input modality combinations. **3)** It is also notable that shared experts are frequently selected among activated experts, suggesting that these shared experts have learned and contain common knowledge that can interact with various modalities, aligning with the motivation behind designing shared experts as a buffer.

**4)** In Figure 3 (c), which shows the token selection ratio from the expert's perspective, it is expected that $BCGI$ is widely chosen by the experts, as this full modality combination is the majority in the ADNI dataset. This combination is frequently sampled through the supporting group, serving as a reference for missing cases. **5)** We also observe that experts select the necessary inputs, such as $B_3, B_4, B_5$, which most often select tokens like $CI$. **6)** In summary, by equipping the router and experts with the knowledge to select the most relevant embedding candidates, missing embeddings are

effectively retrieved to interact with other modalities. This, in turn, boosts performance in downstream tasks by leveraging intra- and inter-modal context and a context-aware routing policy.

### 4.4 ABLATION STUDY

To verify the effectiveness of `MoE-Retriever`, we conducted an extensive ablation study using the ADNI dataset in the $\mathcal{I}+\mathcal{G}+\mathcal{C}+\mathcal{B}$ scenario. Key observations include: **1)** Regarding the core module design in `MoE-Retriever`, involving inter-modal context is crucial as it personalizes the specific observed modality context of each sample. **2)** When designing shared experts ($E_{sh}$), it is important to strike a balance in the number of shared experts. Having too many can deteriorate the acquisition of specialized knowledge required by modality-specific experts. **3)** For modality-specific experts, selecting too few or too many

Table 3: Ablation Study.

| Model Variants | Acc. | F1 |
|---|---|---|
| MoE-Retriever | **64.52**$\pm$2.55 ($\|E_{\mathcal{T}}\|$=8, $\|E_{Sh.}\|$=4, $\|\mathcal{R}\|$=1) | **63.80**$\pm$2.96 |
| w/o Intra-Modal Context | 61.26$\pm$2.33 | 61.80$\pm$1.67 |
| w/o Inter-Modal Context | 60.97$\pm$1.50 | 61.60$\pm$0.78 |
| w/o Context-Aware Routing | 62.34$\pm$1.25 | 63.11$\pm$2.11 |
| $\|E_{\mathcal{T}}\|$=8, $\|E_{Sh.}\|$=1, $\|\mathcal{R}\|$=1 | 60.60$\pm$1.32 | 59.70$\pm$1.26 |
| $\|E_{\mathcal{T}}\|$=8, $\|E_{Sh.}\|$=2, $\|\mathcal{R}\|$=1 | 63.77$\pm$1.35 | 62.92$\pm$0.28 |
| $\|E_{\mathcal{T}}\|$=8, $\|E_{Sh.}\|$=8, $\|\mathcal{R}\|$=1 | 62.98$\pm$0.79 | 62.75$\pm$1.41 |
| $\|E_{\mathcal{T}}\|$=4, $\|E_{Sh.}\|$=4, $\|\mathcal{R}\|$=1 | 63.14$\pm$2.47 | 60.88$\pm$2.21 |
| $\|E_{\mathcal{T}}\|$=16, $\|E_{Sh.}\|$=4, $\|\mathcal{R}\|$=1 | 60.14$\pm$2.97 | 59.91$\pm$1.22 |
| $\|E_{\mathcal{T}}\|$=8, $\|E_{Sh.}\|$=4, $\|\mathcal{R}\|$=2 | 61.14$\pm$1.85 | 61.04$\pm$1.12 |
| $\|E_{\mathcal{T}}\|$=8, $\|E_{Sh.}\|$=4, $\|\mathcal{R}\|$=4 | 60.54$\pm$2.52 | 60.23$\pm$2.71 |
| 2 x Transformer Layer | 63.34$\pm$0.97 | 62.79$\pm$1.31 |
| Sparse MoE Fusion Layer | 62.84$\pm$2.85 | 63.11$\pm$2.25 |

experts can lead to suboptimal results, emphasizing the need for a balanced number, such as eight. **4)** For the router design, utilizing a single router to handle both intra- and inter-modal contexts proved to be sufficient. The more examples it encounters during training, the more knowledge it is able to accumulate. **5)** In the subsequent fusion layer, we experimented with both a vanilla transformer design and a version with the SMoE layer attached. However, no significant performance gain was observed, suggesting that the utilization of SMoE in embedding retrieval was sufficient.

### 4.5 COMPUTATIONAL EFFICIENCY

In Figure 4, we compare the inference time for a single epoch, computational cost, and the number of parameters for each model across different modality configurations in the ADNI dataset. The results show that `MoE-Retriever` outperforms in all three computational dimensions: **1)** Mean Time, **2)** GFLOPs, and **3)** Number of Parameters, thanks to the adoption of the SMoE design. Notably, as the modality combinations increase, the efficiency is maintained, highlighting the advantage of SMoE, which sparsely activates the relevant parameters. This represents a significant step forward in embedding retrieval design.

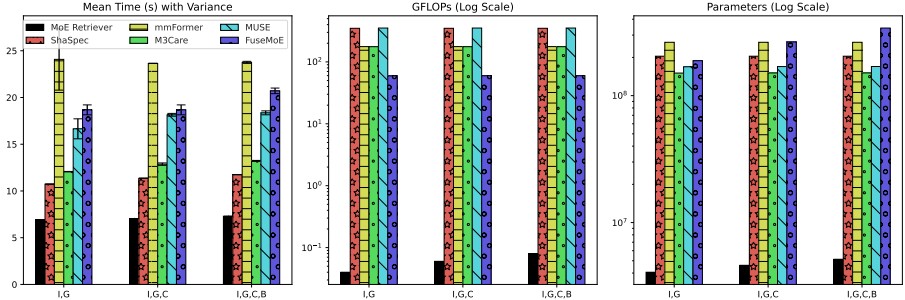

Figure 4: Comparison of computational efficiency of different methods. The left figure displays the averaged inference time for a single epoch of testing data, with error bar showing the variance. The middle plot illustrates the computational cost in GFLOPs (floating-point operations per second divided by $10^9$), while the right figure shows the number of parameters on a logarithmic scale. The FLOPs and GFLOPs are computed using the fvcore package.

## 5 CONCLUSION

In this work, we propose `MoE-Retriever`, a novel framework inspired by the SMoE design that uniquely integrates both intra-modal and inter-modal contexts. By utilizing a modality combination based supporting group for intra-modal context and modality-specific expert which also include shared experts, `MoE-Retriever` effectively selects the most relevant expertes, i.e., embeddings tailored to specific missing modality scenarios. Our extensive experiments on both medical and general machine learning datasets demonstrate that `MoE-Retriever` not only enhances accuracy and robustness in missing modality scenarios but also exhibits scalability and computational efficiency.

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

# A APPENDIX

## A.1 DETAILED PREPROCESSING

We followed the same preprocessing procedure of the ADNI dataset and MIMIC dataset, as described in Flex-MoE (Yun et al., 2024).

### A.1.1 DETAILED DATA PREPROCESSING IN ADNI

**Image Modality** To preprocess the image data, we first applied a correction for magnetic field intensity inhomogeneity to ensure consistency and reliability across MRI images. Next, we used the MUSE (Multiatlas Region Segmentation Utilizing Ensembles of Registration Algorithms and Parameters) method to segment gray matter tissue, the primary focus of this study (Doshi et al., 2016). This technique involves utilizing multiple atlases to extract the most accurate region-of-interest values from the segmented gray matter. Afterward, voxel-wise volumetric maps of tissue regions

were created by spatially aligning skull-stripped images to a template in the Montreal Neurological Institute (MNI) space, using a registration method (Ou et al., 2011).

**Genetic Modality** We obtained SNP (single nucleotide polymorphisms) data from the ADNI 1, GO/2, and 3 studies, and pre-processed it as follows. First, SNP data from these studies were aligned to a unified reference build using Liftover `https://liftover.broadinstitute.org/`, converting all data to NCBI build 37 (UCSC hg19). Next, we aligned strands based on the 1000 Genome Project phase 3, using McCarthy Group Tools `https://www.well.ox.ac.uk/~wrayner/tools/`. Linkage disequilibrium (LD) pruning was then applied with parameters $(50, 5, 0.1)$ to remove highly correlated SNPs, reducing the total SNPs from $565, 989$ to $144, 746$. Imputation was performed on this pruned set using the Michigan Imputation Server `https://imputationserver.sph.umich.edu/index.html#!`, and the resulting SNP data was recoded as $\{0, 1, 2\}$.

**Biospecimen Modality** Biospecimen data was extracted from several ADNI-provided csv files. CSF A$\beta$1-42 and A$\beta$1-40 data were taken from ISOPROSTANE_09May2024.csv, Total Tau and Phosphorylated Tau from UPENNBIOMK_ROCHE_ELECSYS_09May2024.csv, Plasma Neurofilament Light Chain data from batemanlab_20221118_09May2024.csv, and ApoE genotype data from APOERES_09May2024.csv. Numerical data was scaled using a MinMax scaler to a range of -1 to 1, while categorical data was one-hot encoded. For missing values, we imputed the mean for numerical fields and the mode for categorical fields.

**Clinical Modality** Clinical data was extracted from ADNI csv files, including MED-HIST_09May2024.csv, NEUROEXM_09May2024.csv, PTDEMOG_09May2024.csv, REC-CMEDS_09May2024.csv, and VITALS_09May2024.csv. During preprocessing, we excluded the columns 'PTCOGBEG,' 'PTADDX,' and 'PTADBEG,' which contain direct Alzheimer's Disease diagnosis information. Numerical data was scaled using a MinMax scaler (-1 to 1), while categorical data was one-hot encoded. Missing values were imputed by using the mean for numerical columns and the mode for categorical columns.

### A.1.2 DETAILED DATA PREPROCESSING IN MIMIC

**Lab, Notes, Codes Modalities.** For the MIMIC dataset, we use the Medical Information Mart for Intensive Care IV (MIMIC-IV) database, which contains de-identified health data for patients who were admitted to either the emergency department or stayed in critical care units of the Beth Israel Deaconess Medical Center in Boston, Massachusetts24. MIMIC-IV excludes patients under 18 years of age. We take a subset of the MIMIC-IV data, where each patient has at least more than 1 visit in the dataset as this subset corresponds to patients who likely have more serious health conditions. For each datapoint, we extract ICD-9 codes, clinical text, and labs and vital values. Using this data, we perform binary classification on one-year mortality, which foresees whether or not this patient will pass away in a year. We drop visits that occur at the same time as the patient's death.

**Missingness in MIMIC dataset. Code Modality**: This combines diagnosis and procedure data. There are 4 records with missing diagnoses and 1777 with missing procedures. **Note Modality**: Derived from the "text" column of the original CSV file, there are 108 records with missing notes. **Lab Modality**: This presents a more complex scenario, as it includes 2172 different measurements. If we consider all 2172 measurements as potentially missing, then technically, there is no missing data since essential measurements, like heart rate, are consistently collected for each patient. However, if we evaluate the proportion of missing values in the (9003, 2172) matrix, we find that 94.216% of the entries are NaN.

### A.2 MODALITY-SPECIFIC ENCODER SETTINGS

**ADNI Dataset.** For image modality, we used a customized 3D-CNN (Esmaeilzadeh et al., 2018) with hidden dimension 256 as encoder . For genomics, clinical, and biospecimen modalities, we used MLP with hidden dimension 256 as encoder. **MIMIC Dataset.** For all lab, note, and code modalities, we used LSTM with hidden dimension 256 as encoder. **ENRICO Dataset.** For both screenshot image and wireframe image modality, we used VGG11 from torchvision library with hidden dimension size 16 as encoder. **CMU-MOSI Dataset.** For both vision, audio, and text modality, we used Gated Recurrent Unit with hidden dimension 256 as encoder.

## A.3 DIFFERENT NOISE LEVEL OF CMU-MOSI DATASET

| Dataset | Modality | Noise Level | mmFormer | ShaSpec | M3Care | MUSE | FuseMoE | MoE-Retriever |
|---|---|---|---|---|---|---|---|---|
| CMU-MOSI | $\mathcal{V}+\mathcal{A}$ | 0.1 | $51.82_{\pm 1.36}$ | $52.90_{\pm 3.28}$ | $52.62_{\pm 1.59}$ | $51.49_{\pm 1.95}$ | $50.25_{\pm 2.97}$ | $\mathbf{53.19}_{\pm 1.22}$ |
| | | 0.3 | $50.60_{\pm 2.26}$ | $46.61_{\pm 2.36}$ | $47.26_{\pm 2.66}$ | $50.70_{\pm 2.01}$ | $47.46_{\pm 1.51}$ | $\mathbf{53.12}_{\pm 2.34}$ |
| | | 0.5 | $\mathbf{49.95}_{\pm 1.58}$ | $47.39_{\pm 5.97}$ | $42.45_{\pm 3.68}$ | $49.42_{\pm 1.93}$ | $46.91_{\pm 3.44}$ | $49.67_{\pm 3.80}$ |
| | $\mathcal{V}+\mathcal{T}$ | 0.1 | $62.51_{\pm 1.43}$ | $65.75_{\pm 1.39}$ | $69.20_{\pm 0.08}$ | $49.02_{\pm 1.95}$ | $63.49_{\pm 0.98}$ | $\mathbf{66.29}_{\pm 1.99}$ |
| | | 0.3 | $59.78_{\pm 1.10}$ | $62.04_{\pm 1.12}$ | $62.02_{\pm 0.26}$ | $49.16_{\pm 1.91}$ | $55.41_{\pm 3.12}$ | $\mathbf{63.29}_{\pm 2.54}$ |
| | | 0.5 | $57.85_{\pm 0.65}$ | $59.37_{\pm 1.36}$ | $54.63_{\pm 0.76}$ | $48.71_{\pm 1.92}$ | $52.39_{\pm 1.73}$ | $\mathbf{62.49}_{\pm 1.36}$ |
| | $\mathcal{A}+\mathcal{V}$ | 0.1 | $63.54_{\pm 0.73}$ | $66.98_{\pm 0.48}$ | $67.79_{\pm 4.42}$ | $53.00_{\pm 1.94}$ | $69.58_{\pm 0.51}$ | $\mathbf{69.78}_{\pm 2.21}$ |
| | | 0.3 | $61.40_{\pm 1.17}$ | $63.07_{\pm 4.19}$ | $65.77_{\pm 2.69}$ | $46.93_{\pm 1.93}$ | $58.11_{\pm 2.24}$ | $\mathbf{65.27}_{\pm 0.38}$ |
| | | 0.5 | $55.61_{\pm 3.14}$ | $58.06_{\pm 3.65}$ | $46.83_{\pm 5.06}$ | $44.73_{\pm 1.91}$ | $50.86_{\pm 4.65}$ | $\mathbf{62.14}_{\pm 3.17}$ |
| | $\mathcal{V}+\mathcal{A}+\mathcal{T}$ | 0.1 | $64.38_{\pm 1.37}$ | $67.78_{\pm 1.56}$ | $68.29_{\pm 1.24}$ | $49.42_{\pm 1.94}$ | $69.87_{\pm 1.86}$ | $\mathbf{70.36}_{\pm 1.42}$ |
| | | 0.3 | $59.37_{\pm 0.96}$ | $63.76_{\pm 2.56}$ | $63.11_{\pm 5.80}$ | $49.39_{\pm 1.89}$ | $63.97_{\pm 0.88}$ | $\mathbf{65.25}_{\pm 4.06}$ |
| | | 0.5 | $54.41_{\pm 3.78}$ | $55.71_{\pm 2.37}$ | $48.65_{\pm 3.62}$ | $48.86_{\pm 1.91}$ | $47.67_{\pm 3.48}$ | $\mathbf{58.48}_{\pm 1.06}$ |

Table 4: Accuracy in CMU-MOSI dataset across different modalities and noise levels

## A.4 AUROC AND PRAUC RESULTS IN MIMIC DATASET

| Modality | mmFormer | ShaSpec | M3Care | MUSE | FuseMoE | MoE-Retriever |
|---|---|---|---|---|---|---|
| $\mathcal{L}+\mathcal{N}$ | $67.28_{\pm 2.49}$ | $66.08_{\pm 2.87}$ | $50.63_{\pm 1.44}$ | $55.22_{\pm 1.85}$ | $66.05_{\pm 0.98}$ | $\mathbf{68.33}_{\pm 0.50}$ |
| $\mathcal{L}+\mathcal{C}$ | $65.26_{\pm 2.17}$ | $64.61_{\pm 1.18}$ | $50.09_{\pm 0.93}$ | $59.15_{\pm 1.81}$ | $62.53_{\pm 2.44}$ | $\mathbf{67.41}_{\pm 0.57}$ |
| $\mathcal{N}+\mathcal{C}$ | $62.71_{\pm 2.31}$ | $65.61_{\pm 1.43}$ | $51.06_{\pm 2.64}$ | $49.64_{\pm 1.86}$ | $66.61_{\pm 0.65}$ | $\mathbf{65.01}_{\pm 1.01}$ |
| $\mathcal{L}+\mathcal{N}+\mathcal{C}$ | $68.42_{\pm 1.65}$ | $69.27_{\pm 0.14}$ | $50.08_{\pm 0.21}$ | $67.4_{\pm 1.67}$ | $66.65_{\pm 0.78}$ | $\mathbf{69.39}_{\pm 1.08}$ |

Table 5: AUROC Results of MIMIC with different modality combinations

| Modality | mmFormer | ShaSpec | M3Care | MUSE | FuseMoE | MoE-Retriever |
|---|---|---|---|---|---|---|
| $\mathcal{L}+\mathcal{N}$ | $35.20_{\pm 2.94}$ | $34.07_{\pm 2.26}$ | $23.17_{\pm 0.85}$ | $27.03_{\pm 2.12}$ | $33.50_{\pm 1.01}$ | $\mathbf{36.46}_{\pm 0.66}$ |
| $\mathcal{L}+\mathcal{C}$ | $34.07_{\pm 1.46}$ | $33.76_{\pm 0.67}$ | $23.15_{\pm 0.70}$ | $29.8_{\pm 2.27}$ | $32.19_{\pm 0.68}$ | $\mathbf{34.50}_{\pm 1.41}$ |
| $\mathcal{N}+\mathcal{C}$ | $30.97_{\pm 2.89}$ | $34.36_{\pm 1.55}$ | $23.29_{\pm 1.23}$ | $21.12_{\pm 1.34}$ | $\mathbf{35.24}_{\pm 0.34}$ | $33.29_{\pm 0.85}$ |
| $\mathcal{L}+\mathcal{N}+\mathcal{C}$ | $36.54_{\pm 1.24}$ | $36.62_{\pm 1.17}$ | $22.66_{\pm 0.25}$ | $35.23_{\pm 2.54}$ | $34.59_{\pm 1.40}$ | $\mathbf{36.83}_{\pm 0.10}$ |

Table 6: PRAUC Results of MIMIC with different modality combinations

## A.5 HYPERPARAMETER TUNING OF BASELINE MODEL

| Learning Rate | Hidden Dimension | Acc. | F1 | AUROC | PRAUC |
|---|---|---|---|---|---|
| 1e-4 | 64 | 64.58 ± 1.88 | 64.25 ± 1.55 | 70.51 ± 0.43 | 63.86 ± 1.18 |
| | 128 | 65.17 ± 0.80 | 64.87 ± 0.78 | 72.00 ± 2.48 | 64.72 ± 4.57 |
| | 256 | 63.67 ± 0.76 | 63.61 ± 0.73 | 72.77 ± 0.96 | 66.17 ± 2.24 |
| 1e-3 | 64 | 63.92 ± 1.37 | 63.66 ± 1.45 | 71.98 ± 2.33 | 65.93 ± 3.33 |
| | 128 | 64.09 ± 0.07 | 63.59 ± 0.14 | 72.46 ± 1.21 | 67.61 ± 2.29 |
| | 256 | 64.32 ± 0.61 | 64.18 ± 0.54 | 73.33 ± 1.24 | 66.59 ± 3.51 |
| 1e-2 | 64 | 64.51 ± 2.61 | 64.33 ± 2.49 | 72.67 ± 2.10 | 65.29 ± 1.75 |
| | 128 | 65.06 ± 3.25 | 64.63 ± 2.99 | 73.18 ± 2.71 | 66.62 ± 2.82 |
| | 256 | 63.62 ± 0.50 | 63.29 ± 0.54 | 71.81 ± 0.73 | 63.85 ± 1.50 |

Table 7: Results of different hyperparameters of CMU-MOSI Modality $\mathcal{V}+\mathcal{A}+\mathcal{T}$. Learning rate $1e-3$ and hidden dimension 128 are the optimal hyperparameter provided in the ShaSpec paper.

## A.6 GRADIENT CONFLICT IN CMU-MOSI DATASET

In Table 4 of the main paper, we observe that when three modalities ($\mathcal{V}+\mathcal{A}+\mathcal{T}$) are used, the performance for all models does not reach its peak, even compared to using two modalities ($\mathcal{V}+\mathcal{T}$ or $\mathcal{A}+\mathcal{T}$). This highlights an intriguing phenomenon: *adding more modalities does not always guarantee improved performance*. To investigate this from an optimization perspective, we analyze the gradients when all modalities are provided. Specifically, we compute the derivative of the loss with respect to each modality and measure the cosine similarities between modality pairs to detect potential gradient conflicts. Higher cosine similarity indicates positive correlation between gradients, while lower values suggest conflicts.

In Figure 5(a), where dense models such as ShaSpec (Wang et al., 2023) or mmFormer (Zhang et al., 2022b) (leveraging fully connected layers or Transformers) are used in the fusion layer, $\mathcal{V}+\mathcal{A}$ shows positive synergy. However, other pairs, such as $\mathcal{V}+\mathcal{T}$ and $\mathcal{A}+\mathcal{T}$, exhibit less positive interactions, leading to challenges in simultaneously optimizing $\mathcal{V}+\mathcal{A}+\mathcal{T}$ and negatively impacting overall performance. In contrast, as shown in Figure 5(b), sparse models like FuseMoE (Han et al., 2024), which use SMoE in the fusion layer, demonstrate improved synergy due to SMoE's ability to selectively activate the most relevant experts, thereby reducing interference between modalities.

Finally, in our approach (`MoE-Retriever`), as shown in Figure 5(c), SMoE is applied prior to the fusion layer to retrieve missing modalities before the fusion step. This design further enhances the synergy between modalities, resulting in better optimization compared to other baselines, achieving the best performance (65.21 for $\mathcal{V}+\mathcal{A}+\mathcal{T}$). However, it still does not outperform the model's performance when using two modalities (66.13 for $\mathcal{A}+\mathcal{T}$). These findings highlight the importance of addressing gradient conflicts and carefully synergizing modalities, suggesting a promising direction for future research in multi-modal learning.

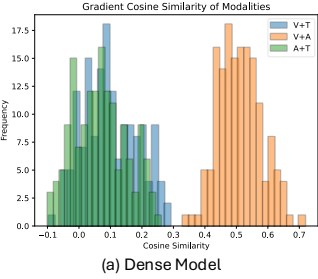 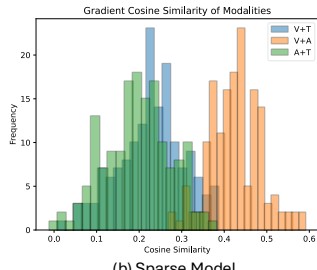 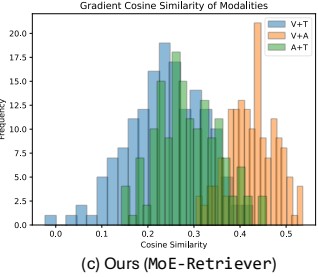

| (a) Dense Model | (b) Sparse Model | (c) Ours (`MoE-Retriever`) |

Figure 5: Gradient similarities of two paired modalities in the CMU-MOSI dataset. (a) Dense model, where the fusion layer is based on a transformer model. (b) Sparse model, where the fusion layer adopts a Sparse Mixture-of-Experts backbone. (c) Ours (`MoE-Retriever`), where SMoE is utilized prior to the fusion layer to retrieve the missing modality. Higher cosine similarity indicates that the gradient operates in a more positive (i.e., same) direction during optimization.

## A.7 Effectiveness of Retrieval from the MIMIC Dataset

To further verify the benefits of the feature retrieval process at the sample (i.e., patient) level, we present a t-SNE plot in Figure 6 using the MIMIC dataset for the one-year mortality prediction task. Specifically, we demonstrate patient-level embeddings, focusing on patients with $\mathcal{LC}$ modalities but missing the $\mathcal{N}$ modality. In Figure 6(a), we observe that before retrieving the $\mathcal{N}$ modality, the embeddings of patients labeled as "alive" are more widely dispersed. In contrast, in Figure 6(b), which incorporates both inter-modal context (i.e., $\mathcal{N}$ embeddings derived from other patients with $\mathcal{LNC}$ modalities) and intra-modal context (i.e., modality-specific embeddings from $\mathcal{L}$ and $\mathcal{C}$), the embeddings are more condensed. By leveraging both contexts, we observe that patient embeddings associated with their respective labels become more similar and compact, positively contributing to the downstream task.

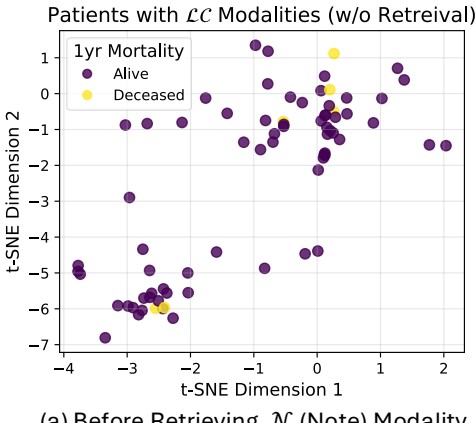 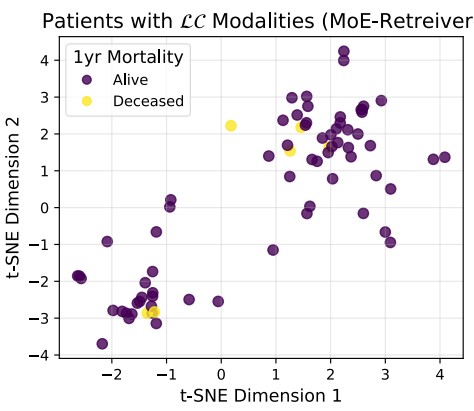

(a) Before Retrieving $\mathcal{N}$ (Note) Modality  (b) After Retrieving $\mathcal{N}$ (Note) Modality

Figure 6: t-SNE plot comparison on the MIMIC dataset with patients having $\mathcal{LC}$ modalities. (a) Before retrieving the $\mathcal{N}$ modality, the embeddings of alive patients are not condensed and appear more scattered. (b) After retrieving the $\mathcal{N}$ modality via `MoE-Retriever`, the embeddings of patients with the alive label become more similar to each other.

