# OpenReview forum: "Sparse MoE as a New Retriever: Addressing Missing Modality Problem in Incomplete Multimodal Data"
_ICLR.cc/2025/Conference — Submitted to ICLR 2025_

### Official Review · Reviewer_nRr5 · 2024-10-28

**Soundness:** 3
**Presentation:** 3
**Contribution:** 2
**Rating:** 8
**Confidence:** 4

**Summary:**

This work attempts to tackle the missing modality problem in multimodal learning. Observing that existing methods do not balance the intra- and inter-modal information when retrieving the representations for missing modalities and the costly computations, the authors tackle these issues by using Sparse MoE. Experiments on general and medical multimodal datasets demonstrate satisfactory performance and thorough ablation studies were performed.

**Strengths:**

•	This work tackles the challenging problem of missing modality, which is one of the most important concerns of multimodal learning.

•	Extensive experiments have been performed on various datasets and recent baseline methods and satisfactory performance is obtained. Ablation study is well-performed.

•	The paper is overall clearly written.

•	Codes and algorithms are available for reproducibility

**Weaknesses:**

· The notion of using MoE to tackle the missing modality problem seems to be well studied by FuseMoE, and the SMoE is also proposed by existing works and extensively applied. More contributions are expected in addition to addressing the balance between intra- and inter-modal information. For instance, there could be more designs on adapting the some into the framework (e.g., how to select the experts based on intra- and inter-modality groupings), given that currently the frameworks just use some with little elaborations on the motivations and modifications. In this sense, I am less convinced that the contribution is enough to meet the bar.

**Minor:**

•	The authors only report the accuracy and F1 scores for primary results, which are highly sensitive to thresholds. Authors are suggested to present the results in AUROC (which is also adapted by MUSE, one of the baselines) in addition to current metrics, to more comprehensively evaluate the performance.

•	The use of English has space of improvement. Some sentences are difficult to follow / less clear. For instance, "lack of specialized knowledge in embedding candidates”"in the abstract is vague and hard for me to follow (i.e., more precise words need to be used for “specialized knowledge”, which is a quite subjective word).

**Questions:**

* Is there any particular advantage on applying the proposed method to medical datasets than general multimodal datasets?
* Is there any specific case from MIMIC (e.g., for a specific diagnosis) to interpret the feature retrieval process of the MoE?
* The CMU-MOSI experiments look interesting. Could the authors also open-source the codes for more information on the implementation?

---

> ### Author Response · Authors · 2024-11-28
> **Response to reviewer nRr5 (1/4)**
>
> We sincerely appreciate Reviewer **nRr5** for highlighting the strengths of MoE-Retriever, such as addressing "one of the most important concerns of multimodal learning," conducting "extensive and satisfactory experiments," and presenting the work in a "clearly written" manner. Below, we provide detailed point-by-point responses to the remaining concerns.
>
> ---
>
> **[W1: Comparison with FuseMoE and More Designs]**
>
> We would first like to clarify the main differences between FuseMoE and our work. While both studies employ Sparse Mixture-of-Experts (SMoE), the fundamental differences lie in how SMoE is utilized in each work. These distinctions can be summarized from the following two perspectives:
>
> 1. **Goal of the Study and Usage of SMoE**
>     Specifically, FuseMoE employs SMoE in the fusion layer, aiming to effectively fuse multimodal information into the same latent space. In contrast, MoE-Retriever utilizes SMoE prior to the fusion layer, with the primary goal of imputing missing modalities while leveraging both intra-modal and inter-modal contexts. This approach enables the effective imputation of missing modality information. Therefore, we respectfully claim that MoE-Retriever is *not* simply an added value on top of FuseMoE, as it does not rely on any backbone design of FuseMoE. The utilization and purpose of SMoE in MoE-Retriever are fundamentally distinct, focusing specifically on **imputation**.
>
> 2. **Router Design**
>     We clarify that router design is not the main focus of our study, unlike prior works that primarily investigate the router design of SMoE [1, 2, 3, 4]. Instead, our study focuses on how to effectively utilize SMoE for **imputation** in the context of missing modalities. Nonetheless, in Section 4.4 (Table 3) of the main paper, we include an ablation study comparing router designs with different numbers of routers to handle both intra- and inter-modal contexts. The results demonstrate that using multiple routers does not outperform a single-router setup, reinforcing the effectiveness of the current router design in MoE-Retriever.
>
>    In summary, rather than advancing router design, our contribution lies in effectively adopting the SMoE framework for the missing modality scenario. Specifically:
>
>    1. **Input Constitution**: Leveraging both intra- and inter-modal contexts as inputs for SMoE.
>    2. **SMoE Goal**: Enabling the router to automatically identify the most relevant experts for imputing missing modality information based on the provided context and modality-specific experts.
>
> Moreover, to facilitate the SMoE design, we focus on refining the inputs to the SMoE, which consist of intra-modal and inter-modal contexts. While the inter-modal context is constructed by simply adding the observed modality-specific embeddings, we have made efforts to further enhance the intra-modal sampling procedure. Specifically, we moved beyond random sampling to explore more sophisticated approaches that could better capture the relevant intra-modal relationships, ultimately improving the retrieval process. Based on similarity rather than randomness, could further improve performance, we implemented a cosine similarity-based sampling procedure.
>
> More specifically:
>
> 1. For a sample with observed modalities G and C, aiming to retrieve modality I, we first define a supporting group containing samples that possess G and C as a sufficient condition and also include I as the missing modality.
> 2. Since all supporting group samples share G and C as common modalities, we concatenate their embeddings and compare them with the embedding of the original sample using cosine similarity.
> 3. The similarity scores are softmaxed to form a probability distribution, which is then used to guide the sampling process. This transforms the sampling into a multinomial distribution where each trial is weighted by the cosine similarity as a sampling probability.
>
> The results of this similarity-based sampling procedure are shown below:
>
> | Accuracy (Mean Time) | MoE-Retriever   | MoE-Retreiver-cos-sim |
> |----------------------|-------------------|-----------------------|
> | I+G         | **61.09 (6.92s)** | 60.97 (123.28s)   |
> | I+G+C        | 63.12 (7.05s)   | **64.08 (129.88s)** |
> | I+G+C+B       | **64.52 (7.29s)** | 64.21 (141.31s)   |
>
> We observe that while the fine-grained approach brings performance gains in certain cases (e.g., I+G+CI+G+C) due to its more careful sampling process, the overall performance remains more robust in the original version.
>
> One important characteristic to note is that leveraging cosine similarity during the training process significantly increases computational complexity. Since parameter updates are performed batch-wise, the cosine similarity scores used for probability sampling must be recalculated in each batch for all possible samples in the supporting group, resulting in a substantial increase in computation time—approximately 20 times higher than the original mean runtime.

---

> ### Author Response · Authors · 2024-11-28
> **Response to reviewer nRr5 (2/4)**
>
> Therefore, while the fine-grained approach may occasionally enhance performance, the trade-off in terms of computational cost is significant. This underscores that the current version of MoE-Retriever strikes a balanced and attractive position between performance and efficiency. Moreover, these findings highlight that the key factor is not merely careful sampling within a pool but, more importantly, how the sampling pool itself is defined—for instance, the modality-combination-based supporting group proves to be crucial for robust performance.
>
> In summary, MoE-Retriever specifically emphasizes incorporating the SMoE design in the missing modality scenario, functioning as a retrieval process rather than solely focusing on effective fusion. Additionally, its balanced design, achieving both strong performance and computational efficiency, further highlights its practical significance. We respectfully argue that the novelty and contributions of our work should not be considered minor elaborations but rather meaningful advancements in the multimodal learning field.
>
> [1] Fan, Dongyang, Bettina Messmer, and Martin Jaggi. "Towards an empirical understanding of MoE design choices." arXiv preprint arXiv:2402.13089 (2024).
> [2] Huang, Quzhe, et al. "Harder Tasks Need More Experts: Dynamic Routing in MoE Models." arXiv preprint arXiv:2403.07652 (2024).
> [3] Dai, Damai, et al. "Stablemoe: Stable routing strategy for mixture of experts." arXiv preprint arXiv:2204.08396 (2022).
> [4] Liu, Tianlin, et al. "Routers in vision mixture of experts: An empirical study." arXiv preprint arXiv:2401.15969 (2024).
>
> ---
>
> **[W2: AUROC Results on CMU-MOSI Dataset]**
>
> We appreciate the reviewer for suggesting AUROC as a useful metric to further evaluate the performance of our model. As a response, during the rebuttal period, we conducted additional experiments on the MIMIC and CMU-MOSI datasets, including varying missing rates (noise levels) for CMU-MOSI to provide a more comprehensive evaluation. The results are summarized below:
>
> **Table 1: AUROC results of MIMIC**
> | **Modality** | **mmFormer** | **ShaSpec** | **M3Care** |  **MUSE**  | **FuseMoE** | **MoE-Retriever** |
> |:------------:|:------------:|:------------:|:------------:|:------------:|:------------:|:-----------------:|
> |   L+N   | 67.28 ± 2.49 | 66.08 ± 2.87 | 50.63 ± 1.44 | 55.22 ± 1.85 | 66.05 ± 0.98 | **68.33 ± 0.50** |
> |   L+C   | 65.26 ± 2.17 | 64.61 ± 1.18 | 50.09 ± 0.93 | 59.15 ± 1.81 | 62.53 ± 2.44 | **67.41 ± 0.57** |
> |   N+C   | 62.71 ± 2.31 | 65.61 ± 1.43 | 51.06 ± 2.64 | 49.64 ± 1.86 | 66.61 ± 0.65 | **65.01 ± 1.01** |
> |   L+N+C  | 68.42 ± 1.65 | 69.27 ± 0.14 | 50.08 ± 0.21 | 67.4 ± 1.67 | 66.65 ± 0.78 | **69.39 ± 1.08** |
>
> **Table 2: AUROC results of CMU-MOSI** **Modality V+A+T**
> | **Missing rate** | **mmFormer** | **ShaSpec** | **M3Care** |  **MUSE**  | **FuseMoE** | **MoE-Retriever** |
> |:----------------:|:------------:|:------------:|:------------:|:------------:|:------------:|:-----------------:|
> |    0.1    | 69.65 ± 1.99 | 75.34 ± 0.70 | 73.33 ± 1.63 | 58.81 ± 2.2 | 77.11 ± 2.06 | **77.26 ± 1.45** |
> |    0.3    | 66.90 ± 1.22 | 72.11 ± 1.60 | 68.17 ± 0.97 | 51.78 ± 2.24 | 72.06 ± 2.71 | **72.94 ± 0.74** |
> |    0.5    | 60.84 ± 4.77 | 63.16 ± 2.46 | 53.88 ± 1.34 | 55.34 ± 2.23 | 56.58 ± 2.86 | **65.94 ± 2.29** |
>
> **Table 3: AUROC Results of CMU-MOSI** **Modality V+A**
> | **Missing rate** | **mmFormer** | **ShaSpec** | **M3Care** |  **MUSE**  |  **FuseMoE**  | **MoE-Retriever** |
> |:----------------:|:------------:|:------------:|:------------:|:------------:|:----------------:|:-----------------:|
> |    0.1    | 57.20 ± 0.68 | 58.05 ± 1.13 | 58.79 ± 2.02 | 53.9 ± 2.31 | **59.45 ± 2.85** |  55.61 ± 1.32  |
> |    0.3    | 51.75 ± 3.70 | 56.50 ± 2.21 | 57.13 ± 1.41 | 51.24 ± 2.3 | **56.75 ± 0.74** |  56.54 ± 0.93  |
> |    0.5    | 53.99 ± 4.14 | 54.21 ± 2.25 | 52.47 ± 3.45 | 52.54 ± 2.29 | **55.41 ± 2.93** |  52.85 ± 1.70  |
>
> **Table 4: AUROC Results of CMU-MOSI** **Modality V+T**
> | **Missing rate** | **mmFormer** | **ShaSpec** |  **M3Care**  |  **MUSE**  | **FuseMoE** | **MoE-Retriever** |
> |:----------------:|:------------:|:------------:|:----------------:|:------------:|:------------:|:-----------------:|
> |    0.1    | 67.94 ± 1.81 | 72.50 ± 1.75 | **72.69 ± 1.18** | 54.42 ± 2.28 | 71.62 ± 0.44 |  71.34 ± 0.70  |
> |    0.3    | 64.74 ± 1.62 | 69.02 ± 1.02 |  67.97 ± 1.90  | 50.18 ± 2.29 | 64.89 ± 1.28 | **70.13 ± 1.29** |
> |    0.5    | 63.03 ± 0.49 | 65.60 ± 2.58 |  49.87 ± 1.14  | 53.53 ± 2.29 | 54.87 ± 1.12 | **68.08 ± 2.20** |

---

> ### Author Response · Authors · 2024-11-28
> **Response to reviewer nRr5 (3/4)**
>
> **Table 5: AUROC Results of CMU-MOSI** **Modality A+T**
> | **Missing rate** | **mmFormer** | **ShaSpec** | **M3Care** |  **MUSE**  | **FuseMoE** |  MoE-Retriever |
> |:----------------:|:------------:|:------------:|:------------:|:------------:|:------------:|:----------------:|
> |    0.1    | 69.82 ± 0.51 | 74.94 ± 0.82 | 69.87 ± 1.52 | 57.89 ± 2.14 | 77.63 ± 0.74 | **77.81 ± 0.51** |
> |    0.3    | 66.02 ± 0.87 | 70.11 ± 3.11 | 69.52 ± 2.92 | 53.51 ± 2.2 | 67.84 ± 0.82 | **72.89 ± 1.10** |
> |    0.5    | 61.25 ± 3.03 | 66.46 ± 2.06 | 57.09 ± 3.50 | 57.33 ± 2.09 | 58.62 ± 6.94 | **69.60 ± 2.61** |
>
> From the above tables, we observe that MoE-Retriever outperforms the baselines in the majority of cases. This aligns with the results presented in the paper using Accuracy and F1 metrics, further demonstrating MoE-Retriever's robust performance, particularly in imbalance scenarios such as the MIMIC dataset.
>
> We have incorporated the above results into our refined version and also included the PRAUC results as suggested by Reviewer **3etg**. We kindly ask the reviewer to refer to the Appendix for a more comprehensive evaluation of the results.
>
> ---
>
> **[W3: English Expression]**
>
> Thank you for pointing out the vague expression in the abstract. What we intended to convey with the phrase "lack of specialized knowledge in embedding candidates" is that current learnable embeddings often lack modality-specific or relevant knowledge that could enhance the imputation process and downstream tasks.
>
> The term "specialized knowledge" is commonly used in SMoE literature, often in the context of experts. However, to make the expression clearer and more reader-friendly, we have revised it in the refined version to "**lack of modality-specific knowledge**" which we believe is more concrete and straightforward.
>
> ---
>
> **[Q1: Advantage of MoE-Retriever on Medical Datasets]**
>
> We would like to clarify that our method is designed to be domain-agnostic and is not specifically tailored for the medical domain. However, its benefits are particularly notable on medical datasets due to the inherent challenges of such data, including more diverse modalities and frequent missingness. In the medical context, missing modalities often occur due to the cost of experiments or specific patient-related factors, making an effective retrieval process critical for better performance.
>
> Empirically, as shown in main paper Table 1 (Medical) and Table 2 (General), the performance gap between MoE-Retriever and the baselines is slightly larger in Table 1. This indicates that in scenarios with more diverse and severe missing modalities, MoE-Retriever can actively address these challenges, providing a significant advantage.
>
> However, we would like to emphasize again that our proposed method is not restricted to the medical domain. It is designed to be generalizable across various domains and scenarios, demonstrating robust performance in a wide range of applications.

---

> ### Author Response · Authors · 2024-11-28
> **Response to reviewer nRr5 (4/4)**
>
> ---
>
> **[Q2: Feature Retrieval in MIMIC Dataset]**
>
> In the MIMIC dataset, the label for each patient is not tied to a specific diagnosis (as in the ADNI dataset) but instead pertains to one-year mortality classification, as mentioned in Appendix A1.2 of the original paper.
>
> To address the reviewer’s inquiry regarding how the retrieval process contributes to performance at the sample (i.e., patient) level, we provide a t-SNE plot in Figure 6 (Appendix 7) using the MIMIC dataset for the one-year mortality prediction task.
>
> The plot can also be accessed via the following link:
>
> - https://anonymous.4open.science/r/moe-retriever-rebuttal-24A8/assets/tsne.png
>
> Specifically, we illustrate patient-level embeddings, focusing on patients with $\mathcal{LC}$ modalities but missing the $\mathcal{N}$ modality. In Figure 6 (a), before retrieving the $\mathcal{N}$ modality, the embeddings of patients labeled as "alive" are more widely dispersed. In contrast, Figure 6 (b), which incorporates both inter-modal context (i.e., $\mathcal{N}$ embeddings derived from other patients with $\mathcal{LNC}$ modalities) and intra-modal context (i.e., modality-specific embeddings from $\mathcal{L}$ and $\mathcal{C}$), shows more condensed embeddings.
>
> By leveraging both intra- and inter-modal contexts, we observe that patient embeddings associated with their respective labels become more similar and compact. This improvement in embedding space positively contributes to the downstream task.
>
> ---
>
> **[Q3: Open-Sourcing the Code for Processing]**
>
> Thank you for your interest in the CMU-MOSI experiment setup. In the experiment, we simulated missing modalities with a probability of 0.3 for each modality independently (and included additional variants during the rebuttal period).
>
> The code for this experiment is available in the following anonymous GitHub repository:
>
> - Link: https://anonymous.4open.science/r/moe-retriever-rebuttal-24A8/dataset.py
>
> ---
>
> We sincerely appreciate the reviewer **nRr5** time and effort in reviewing our paper. If you have any remaining concerns, please feel free to reach out. We would be more than happy to address them and are fully prepared to provide our best responses until the deadline ends.
>
> Best,
> Authors

---

> ### Comment · Reviewer_nRr5 · 2024-11-29
> **Thank you for your response.**
>
> I thank the authors for their detailed response and extensive additional experiments. I think the response well-addresses my concern about the novelty compared to existing MoE methods, the lack of comparable metrics, and the details of implementations. Therefore I have raised my scores.
>
> Besides, I notice that the accuracies and F1s for mmFormer on MIMIC are exactly the same with zero SD across different modality combinations. Could the author briefly explain why this is the case?

---

> ### Author Response · Authors · 2024-11-29
>
> We thank reviewer **nRr5**'s prompt engagement and feedback. We are pleased that our responses have alleviated the reviewer’s concerns regarding the novelty and metrics, and we sincerely appreciate the reviewer’s acknowledgment and consideration of increasing the score.
>
> ---
>
> **[Follow up Question: Consistent Performance of Acc and F1 for mmFormer on MIMIC Dataset]**
>
> We appreciate the detailed question. The consistent accuracy and F1 scores across modality combinations in mmFormer arise partly from its architecture, which emphasizes modality-invariant global features through the modality-correlated Transformer. While mmFormer also incorporates local features via convolutional layers in its hybrid modality-specific encoder, these local features may not be fully captured under imbalanced settings. The reliance on standard cross-entropy loss and zero imputation during training limits the model’s ability to emphasize local context from more informative modalities when some modalities (e.g., $\mathcal{N}$ in MIMIC dataset) are missing. This imbalance leads to a stronger reliance on global, invariant representations, which can smooth out the impact of individual modalities and result in consistent binary predictions across modality combinations, often outputting the same predicted label regardless of the modality combination.
>
> However, although accuracy and F1 scores appear consistent across combinations in Table 1 in main paper, metrics like **AUROC** (`Table 5 in Appendix A.4`) and **PRAUC** (`Table 6 in Appendix A.4`) show differences because they measure the ranking and confidence of predicted probabilities, rather than binary decisions. These metrics reveal that the predicted probabilities vary due to differences in how the model balances local and global information across combinations. When key modalities are missing, the model’s reliance on global representations from the modality-correlated encoder leads to shifts in probability distributions, which are captured by AUROC and PRAUC. Other baselines, which may not focus as heavily on global modality-invariance, demonstrate variability across all metrics, suggesting that mmFormer's robustness to missing modalities comes at the cost of reduced sensitivity to local, modality-specific dynamics. In this regard, treating each modality independently, such as **employing modality-specific experts** responsible for retrieving embeddings for missing modalities, as proposed in MoE-Retriever, can be an attractive remedy in such scenarios.
>
> To clarify this phenomenon further, we will explicitly mention this unique characteristic of mmFormer in the final version to ensure readers fully understand its implications.
>
> ---
>
> Thank you again for your engagement. If you have further concerns, please do not hesitate to let us know.

---

> > ### Comment · Reviewer_nRr5 · 2024-11-29
> >
> > Thanks for the response. I have no further comments.

---

### Official Review · Reviewer_UWni · 2024-11-04

**Soundness:** 2
**Presentation:** 2
**Contribution:** 3
**Rating:** 6
**Confidence:** 4

**Summary:**

This paper introduces MoE-Retriever to address the missing modality problem in medical data. MoE-Retriever leverages intra-modal context by using other samples with the same missing patterns and inter-modal context from available modalities. These contexts are processed through a multi-head attention layer and then aggregated using a sparse mixture-of-experts architecture. Experimental results show that MoE-Retriever improves performance across multiple datasets.

**Strengths:**

1. The approach of leveraging other samples to “impute” missing modalities is intuitive and well-motivated.
2. The use of a sparse mixture-of-experts architecture is reasonable.
3. The experiments are extensive, and the proposed method demonstrates improved performance over existing methods on multiple datasets.

**Weaknesses:**

1. Novelty: The model primarily applies existing techniques, which may limit its novelty.
2. Architecture: MoE-Retriever appears to select support samples randomly. This could lead to inconsistencies if selected samples differ significantly from the input sample. Incorporating a similarity measure might enhance retrieval accuracy.
3. Presentation: The paper’s clarity could be improved, especially in notation, as the numerous superscripts and subscripts can be difficult to follow. Additionally, there is an unusually large margin on page 6.

**Questions:**

See weakness 2.

---

> ### Author Response · Authors · 2024-11-28
> **Response to reviewer UWni (1/2)**
>
> We appreciate Reviewer **UWni** for acknowledging MoE-Retriever’s motivation of "imputing missing modalities" and the "reasonable usage of SMoE." Below, we address the remaining weaknesses point by point.
>
> ---
>
> **[W1: Limited Novelty]**
>
> We sincerely thank the reviewer for highlighting the strengths of our work, particularly the intuitive approach to imputing missing modalities using SMoE. Our main contribution lies not in merely applying existing techniques but in demonstrating their proper and relevant usage in the context of missing modalities in multimodal learning.
>
> In prior works within this domain, such as FuseMoE, SMoE has been effectively utilized due to its efficient router-expert design, but predominantly in the fusion layer. Conversely, for imputing missing modalities, existing approaches often rely on simple learnable features or complete modalities, overlooking the nuanced context of the missing modality.
>
> Our work targets this specific gap by emphasizing the importance of carefully leveraging both intra-modal and inter-modal contexts. Given these contexts, SMoE’s design principle enables the selection of the most relevant experts to retrieve the missing modality. This bridges the gap between missing modality imputation and the efficient utilization of SMoE.
>
> To verify the significance of our proposed modules—**intra-modal context**, **inter-modal context**, and **context-aware routing**—we refer to the ablation study in Table 3 of Section 4.4. The results demonstrate the individual importance of each module and their synergistic effect when integrated.
>
> In this regard, we respectfully argue that our work is novel, as it is the first to address missing modality imputation through the lens of SMoE design principles, where its application is both necessary and natural.
>
> ---
> **[W2: Architecture - Incorporating Similarity Measure]**
>
> We thank the reviewer for suggesting this interesting and promising idea to further enhance our current architecture. As mentioned, in our current intra-modal sampling process, we rely on sampling from a modality-combination-based supporting group. The effectiveness of this approach is demonstrated in the ablation study (Table 3, Section 4.4), where eliminating intra-modal context (i.e., "w/o Intra-Modal Context") negatively impacts overall performance. This highlights the importance of including sampled intra-modal examples.
>
> However, as the reviewer pointed out, our current approach involves random sampling of intra-modal examples, under the assumption that samples sharing similar modality combinations and including the missing modality can positively contribute to the retrieval process. To evaluate whether a more fine-grained sampling strategy, based on similarity rather than randomness, could further improve performance, we implemented a cosine similarity-based sampling procedure.
>
> More specifically:
>
> 1. For a sample with observed modalities G and C, aiming to retrieve modality I, we first define a supporting group containing samples that possess G and C as a sufficient condition and also include I as the missing modality.
> 2. Since all supporting group samples share G and C as common modalities, we concatenate their embeddings and compare them with the embedding of the original sample using cosine similarity.
> 3. The similarity scores are softmaxed to form a probability distribution, which is then used to guide the sampling process. This transforms the sampling into a multinomial distribution where each trial is weighted by the cosine similarity as a sampling probability.
>
> The results of this similarity-based sampling procedure are shown below:
>
> | Accuracy (Mean Time) | MoE-Retriever   | MoE-Retreiver-cos-sim |
> |----------------------|-------------------|-----------------------|
> | I+G         | **61.09 (6.92s)** | 60.97 (123.28s)   |
> | I+G+C        | 63.12 (7.05s)   | **64.08 (129.88s)** |
> | I+G+C+B       | **64.52 (7.29s)** | 64.21 (141.31s)   |
>
> We observe that while the fine-grained approach brings performance gains in certain cases (e.g., I+G+CI+G+C) due to its more careful sampling process, the overall performance remains more robust in the original version.
>
> One important characteristic to note is that leveraging cosine similarity during the training process significantly increases computational complexity. Since parameter updates are performed batch-wise, the cosine similarity scores used for probability sampling must be recalculated in each batch for all possible samples in the supporting group, resulting in a substantial increase in computation time—approximately 20 times higher than the original mean runtime.

---

> > ### Author Response · Authors · 2024-11-28
> > **Response to reviewer UWni (2/2)**
> >
> > Therefore, while the fine-grained approach may occasionally enhance performance, the trade-off in terms of computational cost is significant. This underscores that the current version of MoE-Retriever strikes a balanced and attractive position between performance and efficiency. Moreover, these findings highlight that the key factor is not merely careful sampling within a pool but, more importantly, how the sampling pool itself is defined—for instance, the modality-combination-based supporting group proves to be crucial for robust performance.
> >
> > ---
> >
> > **[W3: Presentation Regarding the Equations]**
> >
> > We appreciate the reviewer’s feedback on the clarity of the equations and understand that they might appear challenging to follow. To address this, we provide a detailed clarification of the two main equations in the paper, including their primary goals, detailed formulations, and explanations of each component:
> >
> >
> >
> > 1. **Clarification of Equation (2)**
> >
> > - `Main Goal`
> >
> >    - To define the supporting group responsible for sampling intra-modal context, taking into account the observed modality combination.
> > - `Formula`
> >
> >    $$G(j \mid \mathcal{T}, mc) = \\{j \in \\{1, 2, \dots, N\\} \mid mc_{j} \in \mathcal{X}(\mathcal{S} \mid \mathcal{T}, mc) \\}$$
> >    $$\text{where} \\; \mathcal{X}(\mathcal{S} \mid \mathcal{T}, mc) = \\{ S \subseteq \mathcal{M} \mid (mc \subseteq S) \land (\mathcal{T} \in S) \\} \\; ∀T∈M,   ∀mc∈MC$$
> >
> > - `Component`
> >
> >    - **$G(j \mid \mathcal{T}, mc)$:** The set of indices (j) of the supporting group where the target modality ($\mathcal{T}$) and modality combination (mcmc) are given. This supporting group is sampled to select intra-modal examples.
> >    - **$mc_{j} \in \mathcal{X}(\mathcal{S} \mid \mathcal{T}, mc) \\}$:** The modality combination of sample j, which must belong to the set of modality combinations $\mathcal{X}$.
> >    - **$\mathcal{X}(\mathcal{S} \mid \mathcal{T}, mc)$:** The set of modality combinations ($\mathcal{S}$) where the target modality ($\mathcal{T}$) and modality combination (mcmc) are given.
> >    - **$(mc \subseteq S) \land (\mathcal{T} \in S)$:** The modality combination \( mc must be a subset of S, and the target modality ($\mathcal{T}$) must also be a subset of S. The operator $\land$ denotes the logical "and" operation. This constraint ensures that the set of modality combinations $\mathcal{X}$ meets the required conditions.
> >
> >
> >
> > 2. **Clarification of Equation (3)**
> >
> > - `Main Goal`
> >   - To retrieve the missing (target) modality using modality-specific experts, leveraging both intra- and inter-modal context inputs.
> >
> > - `Formula`
> >
> >    $$\hat{\mathbf{P}}_{i,\mathcal{T}} = \sum _ {e=1}^{|E|}  \mathcal{R}(\mathbf{x})_e \cdot \mathcal{E}^{\mathcal{T}}_e(\mathbf{x})$$
> >
> >    $$\text{where} \\; \mathbf{x} \in \\{\mathbf{P}^{'}{{i}{\text{intra}}, \mathcal{T}}  \cup  \mathbf{P}^{'}_{i,mc}\\}$$
> >
> >    $$\forall i_\text{intra} \in G(\mathcal{T} \mid mc), \forall \mathcal{T} \in \mathcal{M}, \forall mc \in \mathcal{MC}$$
> >
> > - `Component`
> >
> >    - $\hat{\mathbf{P}}_{i,\mathcal{T}}$:
> >      The predicted retrieved embedding for sample $i$ corresponding to its missing modality $\mathcal{T}$. This is the output of the Sparse Mixture-of-Experts (SMoE) module.
> >    - $\mathcal{E}^{\mathcal{T}}_e(\mathbf{x})$:
> >      The modality-specific expert responsible for the target modality $\mathcal{T}$. Each expert $e$ is specialized for a specific modality and is activated based on the input $\mathbf{x}$.
> >    - $\mathbf{x} \in \\{\mathbf{P}^{'}{{i}{\text{intra}}, \mathcal{T}}  \cup  \mathbf{P}^{'}_{i,mc}\\}$:
> >      The input to the SMoE is the union of intra- and inter-modal context embeddings:
> >      - $\mathbf{P'}_{{i}{\text{intra}}, \mathcal{T}}$: The intra-modal context embedding for sample $i$ related to the target modality $\mathcal{T}$. This embedding is derived by applying Multi-Head Attention (MHA) to the intra-modal context.
> >      - $\mathbf{P'}{i,mc}$: The inter-modal context embedding for sample $i$, representing the combination of observed modality embeddings. This embedding is also derived via MHA applied to $\mathbf{P}_{i,mc}$, where $\mathbf{P}$ is obtained from the modality-specific encoder.
> >
> > For a better reading experience for future readers, we have included a concrete example to illustrate the concept before diving directly into the equation. Additionally, we have refined the equation by adding more detailed explanations of its components to enhance clarity and understanding.
> >
> > ---
> >
> > We sincerely appreciate the reviewer **UWni** time and effort in reviewing our paper. If you have any remaining concerns, please feel free to reach out. We would be more than happy to address them and are fully prepared to provide our best responses until the deadline ends.
> >
> > Best,
> > Authors

---

> > > ### Author Response · Authors · 2024-12-01
> > > **Gentle Reminder to reviewer UWni**
> > >
> > > Dear reviewer **UWni**,
> > >
> > > We are grateful for your time and review on our work. As the discussion period nears its end, we wish to confirm whether our responses have sufficiently clarified and addressed your concerns, which are listed below.
> > >
> > > ---
> > >
> > > - **[W1: Limited Novelty]**
> > > - **[W2: Architecture - Incorporating Similarity Measure]**
> > > - **[W3: Presentation Regarding the Equations]**
> > >
> > > ---
> > >
> > > The refined version of the paper is available in the current PDF, where updates are highlighted in orange. Additional experiments conducted during the rebuttal period, incorporating feedback from other reviewers, can be accessed via the link below:
> > > - **Link: https://anonymous.4open.science/r/moe-retriever-rebuttal-24A8/README.md**
> > >
> > > We are happy to provide additional clarifications before the deadline ends. Please do not hesitate to discuss further concerns.
> > >
> > > Best,
> > > Authors

---

> > > > ### Author Response · Authors · 2024-12-02
> > > > **1 Day Reminder to reviewer UWni**
> > > >
> > > > Dear reviewer **UWni**,
> > > >
> > > > We sincerely appreciate the time and effort you have dedicated to reviewing our work, as well as your insightful comments and constructive feedback. We have carefully revised our paper and incorporated relevant discussions and experiments based on your suggestions. At this stage, we believe all your concerns have been addressed in the rebuttal and the revised version of the paper. However, with the revision deadline fast approaching (**only one day remaining**), we kindly seek your feedback to confirm whether our responses and revisions sufficiently resolve your concerns. If there are any remaining issues, we would greatly value the opportunity to address them promptly to ensure the quality and clarity of our work.
> > > >
> > > > We sincerely hope that **our responses meet your expectations and kindly ask you to consider revisiting your rating**.
> > > >
> > > > Best,
> > > > Authors

---

> ### Comment · Area_Chair_mLP2 · 2024-12-02
> **Please respond to the authors of submission 11331**
>
> Dear reviewer UWni,
>
> We are nearing the end of the discussion period, so please respond to the rebuttal by the authors of submission 11331.
>
> All the other reviewers have weighed in, so we are waiting for your response to make a determination concerning this paper.
>
> Please indicate the extent to which your concerns are addressed and explain your decision to update (or not update) your score.
>
> All the best,
>
> The AC

---

> ### Comment · Reviewer_UWni · 2024-12-03
> **Thanks for the response**
>
> I appreciate the authors' efforts in clarifying the methodology and providing additional experiment results. I have raised my score.
>
> FYI: regarding W2, I think using some algorithms like approximate nearest neighbor search might improve the efficiency.

---

> > ### Author Response · Authors · 2024-12-03
> >
> > Dear reviewer **UWni**,
> >
> > We sincerely appreciate your engagement and are glad that our clarification was clear and helpful. Thank you for acknowledging our efforts and for your decision to raise the score to 6.
> >
> > ---
> >
> > Regarding **[W2: Architecture - Incorporating Similarity Measure]**, we used cosine similarity as a probability for multinomial sampling. The procedure involved:
> > - (1) calculating cosine similarities for a batch of samples and the samples in the supporting group
> > - (2) applying a softmax function to the obtained similarities to derive a multinomial distribution for sampling
> > - (3) sampling based on the resulting probabilities to use them as input for intra-modal context
> >
> > In this regard, as the reviewer suggested, incorporating Approximate Nearest Neighbor (ANN) algorithms such as *Locality-Sensitive Hashing (LSH)*, which is well-suited for the cosine similarity metric, could indeed provide a more efficient alternative for this process. Specifically, LSH could significantly reduce the computational overhead in **step (1) by limiting cosine similarity calculations to only the most relevant samples in the supporting group**. LSH would achieve this by projecting data points into hash buckets using random hyperplanes, such that samples with high cosine similarity are likely to fall into the same bucket. By focusing only on the smaller subset of samples within the same hash bucket, the complexity of similarity calculations can be reduced from $\mathcal{O}(n \times d)$, where $n$ is the number of samples and $d$ is the feature dimension, to $\mathcal{O}(m \times d)$, where $m$ (size of the bucket) is much smaller than $n$. This reduction can make sampling steps more scalable for large datasets.
> >
> > At the same time, it will be important to carefully consider the trade-off between performance and runtime efficiency, as LSH introduces an approximation step that may slightly affect the accuracy of the retrieved samples.
> >
> > ---
> >
> > We deeply appreciate this insightful suggestion and will explore the feasibility of integrating ANN-based methods in our final version. Additionally, we will provide a comprehensive discussion on the impact of fine-grained sampling approaches on the MoE-Retriever’s performance and efficiency, as highlighted in our discussion with reviewer **UWni**.
> >
> >
> > Thank you again for your thoughtful feedback and for providing a valuable perspective.
> >
> > Best,
> > Authors

---

### Official Review · Reviewer_3etg · 2024-11-04

**Soundness:** 3
**Presentation:** 2
**Contribution:** 3
**Rating:** 5
**Confidence:** 3

**Summary:**

The paper aims to address the missing modality problem by retrieving the most relevant embedding for the target(missing) modality. To obtain the relevant embedding, both intra- and inter-modal contexts are employed along with a router. The experimental results on medical datasets and general multimodal datasets demonstrate the effectiveness of the approach.

**Strengths:**

- The motivation of the paper is clear and convincing.
- The method used is reasonable and interesting.

**Weaknesses:**

- As the target of the paper is to address missing modality, it would be more convincing if results regarding different missing ratios could be given for robustness illustration.
- Are there any data statistics regarding the missing modalities for MIMIC?

**Questions:**

- As stated in the paper, baseline models with MIMIC dataset suffer from label imbalance problems. Is there any results using PRAUC?
- For results CMU-MOSI, the results using three modalities (V+A+T) across all models are worse than those with two modalities (A+T). What caused the performance drop?
- For results on CMU-MOSI and ENRICO, the dropping rate is 0.3. Is there an ablation study regarding the robustness against different dropping ratios?
- In Figure 4, MoE-Retriever demonstrated better computational efficiency. However, when the number of modalities scales up, other methods remain similar while MoE-Retriever scales up a bit. Can authors provide more insights or further explanations?
- One additional concerning issue is the resembling of figures and experiment results with the paper [1], which is not cited, mentioned, or discussed at all in the submission. Particularly, experiment results of the FuseMoE baseline reported in Table 1 are exactly the same as that reported in Table 1 and 2 of [1]. Please clarify if the results reported are obtained by the own experiments of the authors, or taken from any other papers. Fig. 3(a) is a modified version of Fig. 2(c) of [1] without any acknowledgment of the sources. Please clarify the source of the figure.

[1] Yun, Sukwon, et al. "Flex-MoE: Modeling Arbitrary Modality Combination via the Flexible Mixture-of-Experts." arXiv preprint arXiv:2410.08245 (2024).

**Details Of Ethics Concerns:**

After my initial review, I came across the NeurIPS 2024 spotlight paper “Flex-MoE: Modeling Arbitrary Modality Combination via the Flexible Mixture-of-Experts". I found that the reviewed article is highly suspicious of overlapping contents with the NeurIPS2024 Spotlight paper " The details are as follows:
1. Figure 3(a) in MoE-Retriever vs Figure 2(b) in Flex-MoE.
2. The results of baseline FuseMoE are the same in these two papers without any reference.

---

> ### Author Response · Authors · 2024-11-28
> **Response to reviewer 3etg (1/4)**
>
> We thank Reviewer **3etg** for taking the time to review our paper and for providing constructive feedback, highlighting the "clear and convincing motivation" and "reasonable method." Below, we provide detailed point-by-point responses.
>
> ---
>
> **[W1 & Q3: Different Missing Ratios of CMU-MOSI & ENRICO]**
> We thank the reviewer for emphasizing the importance of analyzing diverse missing scenarios, as this would provide a more comprehensive evaluation. Beyond the fixed noise ratio of 0.3 used in the paper, we have included additional scenarios with lower noise (0.1) and higher noise (0.5). The performance results on the CMU-MOSI dataset are shown below:
>
> **Table 1:** **Accuracy of** **CMU-MOSI** **Modality V+A**
> | **Missing rate** |  **mmFormer**  | **ShaSpec** | **M3Care** |  **MUSE**  | **FuseMoE** | **MoE Retriever** |
> |:----------------:|:----------------:|:------------:|:------------:|:------------:|:------------:|:-----------------:|
> |    0.1    | 51.82 ± 1.36   | 52.90 ± 3.28 | 52.62 ± 1.59 | 51.49 ± 1.95 | 50.25 ± 2.97 | **53.19 ± 1.22** |
> |    0.3    | 50.60 ± 2.26   | 46.61 ± 2.36 | 47.26 ± 2.66 | 50.7 ± 2.01 | 47.46 ± 1.51 | **53.12 ± 2.34** |
> |    0.5    | **49.95 ± 1.58** | 47.39 ± 5.97 | 42.45 ± 3.68 | 49.42 ± 1.93 | 46.91 ± 3.44 | 49.67 ± 3.80   |
>
> **Table 2:** **Accuracy of** **CMU-MOSI** **Modality V+T**
> | **Missing rate** | **mmFormer** | **ShaSpec** | **M3Care** |  **MUSE**  | **FuseMoE** | **MoE Retriever** |
> |:----------------:|:------------:|:------------:|:------------:|:------------:|:------------:|:-----------------:|
> |    0.1    | 62.51 ± 1.43 | 65.75 ± 1.39 | 69.20 ± 0.08 | 49.02 ± 1.95 | 63.49 ± 0.98 | **66.29 ± 1.99** |
> |    0.3    | 59.78 ± 1.10 | 62.04 ± 1.12 | 62.02 ± 0.26 | 49.16 ± 1.91 | 55.41 ± 3.12 | **63.29 ± 2.54** |
> |    0.5    | 57.85 ± 0.65 | 59.37 ± 1.36 | 54.63 ± 0.76 | 48.71 ± 1.92 | 52.39 ± 1.73 | **62.49 ± 1.36** |
>
> **Table 3:** **Accuracy of** **CMU-MOSI** **Modality A+T**
> | **Missing rate** | **mmFormer** | **ShaSpec** | **M3Care** |  **MUSE**  | **FuseMoE** | **MoE Retriever** |
> |:----------------:|:------------:|:------------:|:------------:|:------------:|:------------:|:-----------------:|
> |    0.1    | 63.54 ± 0.73 | 66.98 ± 0.48 | 67.79 ± 4.42 | 53.0 ± 1.94 | 69.58 ± 0.51 | **69.78 ± 2.21** |
> |    0.3    | 61.40 ± 1.17 | 63.07 ± 4.19 | 65.77 ± 2.69 | 46.93 ± 1.93 | 58.11 ± 2.24 | **65.27 ± 0.38** |
> |    0.5    | 55.61 ± 3.14 | 58.06 ± 3.65 | 46.83 ± 5.06 | 44.73 ± 1.91 | 50.86 ± 4.65 | **62.14 ± 3.17** |
>
> **Table 4:** **Accuracy of** **CMU-MOSI** **Modality V+A+T**
> | **Missing rate** |  **mmFormer**  | **ShaSpec** | **M3Care** |  **MUSE**  | **FuseMoE** | **MoE Retriever** |
> |:----------------:|:----------------:|:------------:|:------------:|:------------:|:------------:|:-----------------:|
> |    0.1    | 64.38 ± 1.37   | 67.78 ± 1.56 | 68.29 ± 1.24 | 49.42 ± 1.94 | 69.87 ± 1.86 | **70.36 ± 1.42** |
> |    0.3    | 59.37 ± 0.96   | 63.76 ± 2.56 | 63.11 ± 5.80 | 49.39 ± 1.89 | 63.97 ± 0.88 | **65.25 ± 4.06** |
> |    0.5    |  54.41 ± 3.78  | 55.71 ± 2.37 | 48.65 ± 3.62 | 48.86 ± 1.91 | 47.67 ± 3.48 | **58.48 ± 1.06**   |
>
> From the results in the tables, we observe that in diverse settings (noise ratios: 0.1, 0.3, 0.5) and across various modality combinations (V+A, V+T, A+T, V+A+T), MoE-Retriever demonstrates strong and robust performance in most cases. Furthermore, as the noise ratio increases—indicating more severe missing scenarios—the performance gap between MoE-Retriever and the baselines widens in most cases (as shown in Tables 2, 3, and 4).
>
> This robustness is attributed to the imputation process in MoE-Retriever, which leverages both intra-modal and inter-modal contexts. In contrast, the baselines either rely on simple imputation techniques or focus solely on modality fusion while leveraging only the observed modalities.
>
> ---
> **[W2: Data Statistics of MIMIC Dataset]**
>
> We thank the reviewer for inquiring about the missingness statistics of the MIMIC dataset. Below are the details regarding missingness across the 9,003 patient records:
>
> - **Code Modality:** This modality combines diagnosis and procedure data. There are 4 records with missing diagnoses and 1,777 records with missing procedures.
> - **Note Modality:** Derived from the "text" column in the original CSV file, there are 108 records with missing notes.
> - **Lab Modality:** This modality includes 2,172 different measurements, presenting a more complex missingness scenario. If all 2,172 measurements are considered, there is technically no missing data since critical measurements, such as heart rate, are consistently collected for each patient. However, when evaluating the proportion of missing values in the 9,003×2,1729,003 \times 2,172 matrix, we find that **94.216%** of the entries are NaN.
>
> We hope this clarifies the missingness statistics of the MIMIC dataset. For better transparency, these statistics have been included in Appendix 1.2 of the refined version.

---

> ### Author Response · Authors · 2024-11-28
> **Response to reviewer 3etg (2/4)**
>
> ---
>
> **[Q1: Results of PRAUC on the MIMIC Dataset]**
>
> As the reviewer noted, the use of PRAUC is indeed a valuable metric, particularly for addressing the label imbalance issue in the MIMIC dataset. We have included the PRAUC results for the baseline models and the MoE-Retriever in the table below:
>
> | **Modality** | **mmFormer** | **ShaSpec** | **M3Care** |  **MUSE**  |  **FuseMoE**  | **MoE Retriever** |
> |:------------:|:------------:|:------------:|:------------:|:------------:|:----------------:|:-----------------:|
> |   L+N   | 35.20 ± 2.94 | 34.07 ± 2.26 | 23.17 ± 0.85 | 27.03 ± 2.12 |  33.50 ± 1.01  | **36.46 ± 0.66** |
> |   L+C   | 34.07 ± 1.46 | 33.76 ± 0.67 | 23.15 ± 0.70 | 29.8 ± 2.27 |  32.19 ± 0.68  | **34.50 ± 1.41** |
> |   N+C   | 30.97 ± 2.89 | 34.36 ± 1.55 | 23.29 ± 1.23 | 21.12 ± 1.34 | **35.24 ± 0.34** |  33.29 ± 0.85  |
> |   L+N+C  | 36.54 ± 1.24 | 36.62 ± 1.17 | 22.66 ± 0.25 | 35.23 ± 2.54 |  34.59 ± 1.40  | **36.83 ± 0.10** |
>
> From the results in the table, we observe that MoE-Retriever outperforms the baseline models in the majority of cases. We attribute this performance gain to MoE-Retriever's ability to effectively combine intra-modal and inter-modal contexts, which helps to supplement minority class data. Additionally, its use of a Sparse Mixture-of-Experts (SMoE) router ensures that only the most relevant experts are activated, leading to robust embeddings and higher PRAUC, even in the presence of class imbalance.
>
> This experiment has been added to the refined version in Appendix A.5.
>
> ---
>
> **[Q2: Cause of Performance Drop in the CMU-MOSI Dataset]**
>
> We appreciate the reviewer for raising this issue. This is a fairly common challenge observed across other baselines as well. In all baselines, compared to the V+A+T modality combinations, the best performance is often achieved with only two modalities. For instance, models like mmFormer, ShaSpec, and M3Car perform best with A+T, while MUSE and FuseMoE achieve optimal performance with V+T.
>
> We posit that in multimodal scenarios, including more modalities does not always guarantee an improvement in overall performance. While additional modalities may provide more information, they can also introduce unwanted noise and challenges such as modality interference, leading to gradient conflicts. Addressing such interference is an active area of research in multimodal learning [1, 2, 3], but it is not the primary focus of this paper. Our study centers on how to effectively retrieve missing modalities in multimodal scenarios by leveraging intra- and inter-modal contexts with an SMoE design.
>
> Nonetheless, during the rebuttal period, we investigated why adding more modalities does not always improve performance from an optimization perspective. To analyze this, we examined the gradients when all modalities are included. Specifically, we computed the derivative of the loss with respect to each modality and measured the cosine similarities between modality pairs to detect potential gradient conflicts. Higher cosine similarity indicates a positive correlation between gradients, while lower values suggest conflicts.
>
> We tracked the median cosine similarity values during training for the following configurations:
>
> 1. **Dense Models**: Models with a Transformer-based fusion layer, such as mmFormer.
> 2. **Sparse Models**: Models with an SMoE-based fusion layer, like MoE-Retriever.
>
> The numerical results are presented below:
>
> |        | cos_sim($\frac{\partial L}{\partial \mathcal{V}}$, $\frac{\partial L}{\partial \mathcal{T}}$) | cos_sim($\frac{\partial L}{\partial \mathcal{V}}$, $\frac{\partial L}{\partial \mathcal{A}}$) | cos_sim($\frac{\partial L}{\partial \mathcal{A}}$, $\frac{\partial L}{\partial \mathcal{T}}$) | Mean ($\uparrow$) / Variance ($\downarrow$) |
> |---------------|:-------------:|:-------------:|:-------------:|:-------:|
> | Dense Model |   0.091   |   0.506   |   0.066   | 0.221 / 0.061 |
> | Sparse Model |   0.231   |   0.426   |   0.199   | 0.285 / 0.015 |
> | MoE-Retriever |   0.239   |   0.425   |   0.281   | **0.315** / **0.009** |
>
> For a more comprehensive histogram plot, please refer to the link below or Appendix A.6 in the refined version:
>
> - Link: https://anonymous.4open.science/r/moe-retriever-rebuttal-24A8/assets/image-4.png

---

> ### Author Response · Authors · 2024-11-28
> **Response to reviewer 3etg (3/4)**
>
> From the experiment, we observe that in dense models, while the cosine similarities are higher for the combination of V+A, other combinations, such as V+T and A+T, suffer from relatively low cosine similarities. This imbalance can negatively affect the overall optimization process.
>
> In contrast, using sparse models that leverage SMoE in the fusion layer (e.g., FuseMoE) mitigates this issue by achieving a higher average cosine similarity with lower variance. Furthermore, by incorporating SMoE in the retrieval process of embeddings, which is the core motivation of MoE-Retriever, we achieve the highest average cosine similarities across modalities with the lowest variance. This enables MoE-Retriever to deliver the best performance among all baselines when using three modalities.
>
> These findings underscore the importance of addressing gradient conflicts through SMoE utilization, particularly in its role as a retriever of embeddings, thereby optimizing synergy among modalities. This suggests a promising direction for future research in multimodal learning.
>
> We will include this experiment in the refined version and thank the reviewer for raising this phenomenon, fostering further discussion and progress within the community.
>
> [1] Liu, Kuan, et al. "Learn to combine modalities in multimodal deep learning." *arXiv preprint arXiv:1805.11730* (2018).
> [2] Peng, Jie, et al. "Sparse moe as a new treatment: Addressing forgetting, fitting, learning issues in multi-modal multi-task learning." (2023).
> [3] Yu, Shoubin, Jaehong Yoon, and Mohit Bansal. "Crema: Multimodal compositional video reasoning via efficient modular adaptation and fusion." *arXiv preprint arXiv:2402.05889* 1 (2024).
>
> ---
>
> **[Q4: Computational Efficiency of MoE-Retriever]**
>
> In Figure 4, all baseline models experience an increase in computational cost due to the inclusion of modality-specific encoders and their respective fusion layers. However, the reason these increases appear visually similar is the relatively larger computational scale of the baseline models compared to MoE-Retriever.
>
> Below is the actual table used to generate Figure 4:
>
> |   Model   | Modality Combination | Mean Time (s) |   FLOPs   | Parameters |
> |:-------------:|:--------------------:|:-------------:|:---------------:|:-----------:|
> | MoE Retriever | I+G         |   6.92   |  **35,685,530**  | **4,061,091** |
> |        | I+G+C        |   7.05   |  **55,088,051**  | **4,605,091** |
> |        | I+G+C+B       |   7.29   |  **75,530,411**  | **5,138,339** |
> |  ShaSpec  | I+G         |   10.73   | 349,142,312,587 | 203,220,003 |
> |        | I+G+C        |   11.36   | 349,146,731,898 | 204,217,635 |
> |        | I+G+C+B       |   11.75   | 349,151,151,209 | 204,524,579 |
> |  mmFormer  | I+G         |   24.08   | 176,397,734,272 | 223,998,499 |
> |        | I+G+C        |   23.65   | 176,469,346,688 | 241,890,851 |
> |        | I+G+C+B       |   23.76   | 176,554,973,568 | 263,286,819 |
> |   M3Care  | I+G         |   12.07   | 176,104,367,488 | 150,641,700 |
> |        | I+G+C        |   12.87   | 176,105,862,528 | 150,899,108 |
> |        | I+G+C+B       |   13.22   | 176,106,666,880 | 150,983,844 |
> |   MUSE   | I+G         |   16.65   | 352,379,836,408 | 168,867,620 |
> |        | I+G+C        |   18.15   | 352,398,078,751 | 169,448,484 |
> |        | I+G+C+B       |   18.36   | 352,413,787,718 | 169,684,004 |
> |  FuseMoE  | I+G         |   18.68   | 59,524,507,708 | 188,493,251 |
> |        | I+G+C        |   18.68   | 59,737,697,852 | 264,680,387 |
> |        | I+G+C+B       |   20.71   | 59,761,718,332 | 340,929,475 |
>
> From the table, we observe that baseline models such as ShaSpec, mmFormer, and M3Care, which appear to have relatively stable computational efficiency, actually experience increases in the number of parameters and GFLOPs as the number of modalities increases. However, due to the inherent complexity of their fusion layers—often involving significant stacking of transformer layers—the initial computational cost and parameter count for these baselines are already high when only two modalities are used. Consequently, the relative increase in computational efficiency appears small as additional modalities are incorporated.

---

> > ### Author Response · Authors · 2024-11-28
> > **Response to reviewer 3etg (4/4)**
> >
> > In contrast, MoE-Retriever exhibits a relatively higher increase in GFLOPs, likely due to:
> >
> > 1. The use of multi-head attention in its design.
> > 2. The inclusion of additional experts to handle imputation for missing modalities.
> >
> > However, the increase in the number of parameters for MoE-Retriever remains marginal compared to baselines like mmFormer and FuseMoE. This efficiency is achieved through its SMoE design, which selectively activates only the relevant experts based on intra- and inter-modal contexts.
> >
> > Overall, while an increase in computational cost is observed, on an absolute scale, MoE-Retriever demonstrates significantly reduced running time and a much smaller parameter count (e.g., 2.5% of ShaSpec or 1.5% of FuseMoE) while delivering superior performance compared to all baselines. This highlights MoE-Retriever as a computationally efficient and powerful model.
> >
> >
> >
> > ----
> >
> > **[Q5 & Ethical Concerns: Appropriate Mentioning and Citation]**
> >
> > We thank the reviewer for pointing out the relevance of the Flex-MoE paper to our work. As mentioned, the use of the multi-modal ADNI dataset in the Flex-MoE paper draws motivation from MoE-Retriever. However, we would like to clarify the following points:
> >
> > 1. **Figure Reference:**
> >     We did not directly copy or paste their figure. Instead, we referred to the Venn diagram structure from their paper as an effective backbone to illustrate the complex multi-modal data structure. To emphasize the dependency on the "clinical" modality, we introduced a dotted-line representation, which is distinct from the original diagram.
> > 2. **Baseline Results Reference:**
> >     We did not reference all the results from Flex-MoE's use of the ADNI dataset. Specifically, we referred only to the baseline FuseMoE results on the ADNI and MIMIC datasets. This was necessary because our implementation of FuseMoE, based on the author's code, consistently produced high variance and inconsistent results, especially on these two datasets. As a result, we partially referred to Flex-MoE's results for the FuseMoE baseline.
> >
> > Given the motivations derived from the figure and the use of FuseMoE results, we acknowledge the need for proper citation. During the submission period (ended Oct 01), however, the preprint version of the Flex-MoE paper was not publicly available (submitted Oct 10), which prevented us from citing it. Now that the arXiv version has been published, we have included proper citations in the revised version, particularly referencing the data statistics in Figure 3(a) and the baseline performance of the ADNI dataset in Table 1. For clarity, these citations have been highlighted in orange text in the revised version.
> >
> > ---
> >
> > We sincerely appreciate the reviewer **3etg** time and effort in reviewing our paper. If you have any remaining concerns, please feel free to reach out. We would be more than happy to address them and are fully prepared to provide our best responses until the deadline ends.
> >
> > Best,
> > Authors

---

> > > ### Comment · Reviewer_3etg · 2024-11-29
> > >
> > > Thanks to the authors for providing detailed responses on data statistics, computational efficiencies, different missing ratios, AUPRC results, and gradient conflict details. I have some follow-up comments I would like to share:
> > >
> > > 1. The ethical concern has been addressed with proper citation and therefore I have unflagged it. However, based on the work of Flex-MoE, it seems to me the novelty of MOE-Retriever (the submitted paper) is limited, especially regarding the scope (missing modality in multimodal setting) and the methodology. It would be beneficial if the authors could clarify the differences.
> > >
> > > 2. The authors have attributed the performance drop to gradient conflicts. I am wondering why combining the two most "cooperative" modalities (V+A) archives much lower performance compared to less "cooperative" combinations (V+T, A+T).

---

> ### Author Response · Authors · 2024-12-01
> **Follow up Response to reviewer 3etg (1/2)**
>
> We are pleased to hear that our detailed response has addressed most of reviewer **3etg**’s concerns. For the remaining follow-up questions, we would like to provide our responses below.
>
> ---
>
> **[Follow-up Q1: Comparison with Flex-MoE]**
>
> Thank you for raising this insightful question. We would like to clarify the differences between **Flex-MoE** and **MoE-Retriever** from the following two perspectives: `(a) focus of work` and `(b) methodology`. Additionally, we explore the potential for `(c) positive synergy ` between the two approaches.
>
> - `(a) Focus of Work`
> While both works address the challenge of missing modalities in multi-modal learning—an actively explored field—we respectfully highlight that the problem formulation and the proposed solutions are fundamentally different.
>
>   1. **Flex-MoE** focuses on effectively modeling various modality combinations, particularly in the fusion layer. It achieves this by replacing the FFN layer in a Transformer block with a SMoE layer in the fusion layer. Although it introduces a Missing Modality Bank as a remedy, this component is essentially a learnable bag-of-embeddings approach that *does not consider contextual factors*, such as which observed samples might benefit the sample with missing modalities or how the observed modalities themselves contribute to imputation.
>
>   2. **MoE-Retriever**, on the other hand, emphasizes the careful imputation of relevant embeddings for missing modalities by leveraging intra- and inter-modal contexts. These contexts, which are absent in Flex-MoE, allow MoE-Retriever to generate contextually-aware imputed embeddings before proceeding to the fusion step. For the fusion, MoE-Retriever employs a standard Transformer layer to aggregate multi-modal inputs.
>
>   In summary, **Flex-MoE prioritizes effective fusion techniques**, while **MoE-Retriever focuses on robust imputation methods**.
>
> - `(b) Methodology`
> The methodologies of Flex-MoE and MoE-Retriever reflect their distinct objectives:
>
>   1. **Flex-MoE**:
>    - *Missing Modality Bank*: Provides learnable embeddings derived from observed modality pairs and their corresponding missing modalities.
>    - *Generalized Router*: Utilizes fully observed samples to train modality-combination-based experts in SMoE with generalized knowledge.
>    - *Specialized Router*: Uses few observed samples to specialize experts for each corresponding modality combination.
>
>   2. **MoE-Retriever**:
>    - *Intra-Modal Context*: Defines a supporting group of samples sharing similar observed modalities and missing modalities, capturing intra-modal similarities.
>    - *Inter-Modal Context*: Leverages observed modalities within each sample to provide personalized semantic information.
>    - *Context-Aware Routing Policy*: Based on intra- and inter-modal inputs, the router selects top-k missing modality-specific experts to retrieve the most relevant embeddings.
>
>   In summary, **Flex-MoE prioritizes new routing policies for SMoE to model arbitrary modality combinations**, whereas **MoE-Retriever designs necessary inputs (intra- and inter-modal contexts) for SMoE to retrieve contextually-relevant features for missing modalities**.
>
> - `(c) Exploring Synergies`
> Beyond comparison, we explored potential synergies between Flex-MoE and MoE-Retriever to investigate whether their approaches can complement each other. Specifically, MoE-Retriever was used to generate imputed embeddings for missing modalities, which were then fed into Flex-MoE's router design for fusion. Below are the results of this combination on the CMU-MOSI dataset:
> | Modalities | Flex-MoE       | MoE-Retriever   | MoE-Retriever + Flex-MoE      |
> |:----------:|----------------|-----------------|--------------------------------|
> | V + A      | 46.43 ± 1.58   | 53.12 ± 2.26    | **56.68 ± 2.12** (+6.70%) |
> | V + T      | 62.61 ± 2.70   | 65.74 ± 0.55    | **66.04 ± 1.83** (+0.46%) |
> | A + T      | 61.27 ± 2.79   | 66.13 ± 0.69    | **66.44 ± 1.84** (+0.47%) |
> | V + A + T  | 60.71 ± 2.88   | 65.21 ± 2.72    | **66.76 ± 2.01** (+2.38%) |
>
>
>   From these results, we observe:
>   1. **Flex-MoE alone does not outperform MoE-Retriever**, emphasizing the importance of careful imputation of missing modalities, particularly by considering intra- and inter-modal contexts.
>   2. **The combination of MoE-Retriever and Flex-MoE surpasses the performance of either method alone**, demonstrating that using MoE-Retriever for imputation followed by Flex-MoE for fusion creates a strong baseline. It not only imputes missing modalities effectively but also leverages available modality combinations through expert design.  This shows MoE-Retriever’s generalizability as an imputer capable of complementing advanced fusion techniques.
>
> To enhance clarity for future readers, we will ensure that the comparison and the potential synergy of MoE-Retriever with existing fusion methods being included in the final version.

---

> ### Author Response · Authors · 2024-12-01
> **Follow up Response to reviewer 3etg (2/2)**
>
> ---
>
> **[Follow-up Q2: Cooperativeness and Performance on V+A]**
> Thank you for highlighting this important and insightful question. We would like to emphasize a significant claim that the multi-modal learning field should carefully consider:
>
> > **"Although higher cosine similarity can facilitate faster convergence and improve the optimization process, it does not always guarantee better downstream performance."**
>
> This is because, in addition to pairwise relative similarities, the absolute modality-specific information quality and its contribution to the task must also be taken into account. Now, specifically, in the CMU-MOSI dataset [1], preprocessing [2] treats the **text (T)** modality as the central modality. Specifically, the dataset is preprocessed to ensure temporal alignment among the modalities (vision (V), audio (A), and text (T)) based on word-level granularity. As stated in [2]:
>
> > *"To reach the temporal alignment between different modalities, we choose the granularity of the input to be at the level of words."*
>
> This preprocessing implies that the dataset is heavily centered on the text modality, resulting in an uneven distribution of information across modalities.
>
> The preprocessed `feature shapes` for each modality are as follows:
> - **Vision (V):** (2199, 50, 20)
> - **Audio (A):** (2199, 50, 5)
> - **Text (T):** (2199, 50, 300)
>
> Here, 2199 is the number of samples, 50 represents the number of timestamps, and 20, 5, and 300 are the number of features for vision, audio, and text, respectively. This shows that the text modality has **15×** and **60×** more features than the vision and audio modalities, respectively, making it likely to dominate the downstream performance. However, even with fewer features, individual modalities like vision and audio can significantly contribute to the task if their feature-label relevance is high.
>
> To validate this, we calculated the `feature-label correlation` for each feature (averaged globally across timestamps). The statistics (Max, Mean, Median, Normalized Sum of correlations) for each modality are as follows:
>
> | Modality | Max       | Mean      | Median    | Normalized Sum |
> |----------|-----------|-----------|-----------|----------------|
> | V   | 0.094     | -0.003    | **0.023** | -0.003         |
> | A    | 0.074     | -0.053    | -0.067    | -0.052         |
> | T    | **0.214** | **0.009** | 0.014     | **0.009**      |
>
> For a more comprehensive view of the correlation distributions for each modality, please refer to the plot linked below:
> - Link: **https://anonymous.4open.science/r/moe-retriever-rebuttal-24A8/assets/modality_correlation_plot.png**
>
> As a result, due to the dominance of the text modality, Table 2 in the main paper shows that when the central modality (T) is missing (i.e., V+A), performance is the lowest. This occurs even though the cosine similarity of gradients is the highest, likely because the smaller feature dimensions in vision and audio result in gradients that align more closely pairwise.
>
> These observations suggest that when evaluating modality contributions, it is crucial to consider not only **cosine similarity between gradients** but also the **modality-specific characteristics**, such as feature-label correlation and the amount of information each modality contributes.
>
>
> [1] Liang, Paul Pu, et al. "Multibench: Multiscale benchmarks for multimodal representation learning." Advances in neural information processing systems 2021.DB1 (2021): 1.
>
> [2] Liang, Paul Pu, Ruslan Salakhutdinov, and Louis-Philippe Morency. "Computational modeling of human multimodal language: The mosei dataset and interpretable dynamic fusion." First Workshop and Grand Challenge on Computational Modeling of Human Multimodal Language. Vol. 113. 2018.
>
> ---
>
> We sincerely appreciate reviewer **3etg**’s insightful follow-up feedback and hope that the above responses address the remaining concerns. If there are any further questions or clarifications needed, please do not hesitate to reach out to us before the deadline.
>
> Best,
> Authors

---

> > ### Comment · Reviewer_3etg · 2024-12-02
> >
> > I appreciate the authors' efforts in clarifying the methodology and providing comparisons with the related work Flex-MoE. I have raised my score from 3 to 5.

---

> > > ### Author Response · Authors · 2024-12-02
> > >
> > > Dear reviewer **3etg**,
> > >
> > > We deeply appreciate the time and effort you have devoted to reviewing our paper, as well as your thoughtful engagement throughout the rebuttal process. We are especially grateful that our responses alleviated your concerns regarding the comparison with Flex-MoE and that you considered raising the score from 3 to 5.
> > >
> > > ---
> > >
> > > Over the course of the rebuttal, we had the opportunity to address eight topics in detail:
> > >
> > > - 1. **[W1 & Q3: Different Missing Ratios of CMU-MOSI & ENRICO]**
> > > - 2. **[W2: Data Statistics of the MIMIC Dataset]**
> > > - 3. **[Q1: Results of PRAUC on the MIMIC Dataset]**
> > > - 4. **[Q2: Cause of Performance Drop in the CMU-MOSI Dataset]**
> > > - 5. **[Q4: Computational Efficiency of MoE-Retriever]**
> > > - 6. **[Q5 & Ethical Concerns: Appropriate Mentioning and Citation]**
> > > - 7. **[Follow-up Q1: Comparison with Flex-MoE]**
> > > - 8. **[Follow-up Q2: Cooperativeness and Performance on V+A]**
> > >
> > > ---
> > >
> > > We have done our utmost to address your concerns comprehensively, and we are delighted that the explanations provided have been helpful.
> > >
> > > Given the constructive progress made through our dialogue, we humbly and respectfully request your *consideration of further raising the score to 6*, which would place the paper within the acceptance range. We believe that our work, which introduces the **`pioneering use of SMoE for missing modality imputation`**, has the potential to make a meaningful contribution to the field of multimodal learning.
> > >
> > > If there are additional concerns or reservations that you feel were not fully addressed, we would be more than happy to further clarify and provide additional evidence in the remaining days. Your support at this stage would mean a great deal to us and help ensure that this contribution reaches the broader community.
> > >
> > > Thank you again for your thoughtful engagement and for considering our request.
> > >
> > > Best,
> > > Authors

---

> > > > ### Author Response · Authors · 2024-12-03
> > > > **Final Reminder for Reviewer 3etg's Consideration**
> > > >
> > > > Dear reviewer **3etg**,
> > > >
> > > > We once again thank you for your thoughtful engagement and for the valuable feedback you have provided throughout the review process. Your detailed comments and follow-up questions have been instrumental in helping us improve the clarity and quality of our paper. We are sincerely grateful for your support and for raising your score, which we deeply appreciate.
> > > >
> > > > As the review period enters its final hours, we humbly ask if you might consider *revisiting your score once more*. With your encouragement and the improvements made during the rebuttal, MoE-Retriever now stands in a **balanced position among the ratings, and your final support could significantly impact its chances of acceptance**. If the reviwer find our responses and revisions satisfactory and are open to further supporting our work, **even a small additional increase** in score would mean a great deal to us.
> > > >
> > > > All in all, we fully respect the effort and time you have already dedicated to reviewing our submission and would like to sincerely thank you again for the insightful discussions and constructive feedback that have significantly enhanced our work.
> > > >
> > > > Best,
> > > > Authors

---

### Official Review · Reviewer_L5fQ · 2024-11-05

**Soundness:** 3
**Presentation:** 1
**Contribution:** 2
**Rating:** 3
**Confidence:** 4

**Summary:**

This paper proposes a model called MoE-Retriever to retrieve embeddings for missing modalities in multimodal machine learning. Both intra-modal and inter-modal contexts are considered in the embedding retrieval process. The former considers other data samples with the target modality to be retrieved by proposing a supporting group and the later considers the available modalities from the same data sample. Empirical evaluation shows improvement over existing methods on several public datasets.

**Strengths:**

- The problem of missing modality in the context of multimodal machine learning is significant and worth investigating.
- The idea of considering intra-modal and inter-modal contexts for embedding retrieval is interesting and important.
- The empirical evaluation shows improvement over existing methods.

**Weaknesses:**

The major weaknesses are the limited novelty and unclear presentation. Detailed comments are as follows.

1. The novelty seems limited. The proposed MoE is largely developed based on FuseMoE with the same router design. The added value on top of FuseMoE is mainly the intra- and inter-modal contexts.
2. The intra-modal context section, especially Eq. (2), looks very confusing to me. $\mathcal{M}$ is a set of all modalities, $\mathcal{X}(\mathcal{T}|mc)$ is a set of subsets of $\mathcal{M}$, why the integer $j$ is an element of it? What does $S\wedge \mathcal{T}$ mean? Or is it supposed to be $S\cap \mathcal{T}$? Is $S$ a set? why $S$ is a subset of $\mathcal{M}$ and at the same time $S\wedge\mathcal{T}$ is an element of $S$? After carefully reading Eq. (2) and the descriptions below it, I still find it difficult to understand how the intra-modal context is extracted and used.
3. The inter-modal context is not described in detail. Particularly, it is argued that the missing modality of imaging may indicate an early stage of AD, but it is not clear how the proposed inter-modal context can help address this issue.
4. In Eq. (3), are $\mathbf{P}^\prime$, predicted retrieved embedding, matrices? What does it mean to take the union between two matrices?
5. It is not clear how each expert is parameterized. What is the model used for each expert?
6. What does "input token" mean? Is it one data sample or one particular feature from a data sample?
7. For implementation, using optimal hyperparameter settings in the original paper does not necessarily guarantee a fair comparison since the baselines may not be tuned in the datasets used in this work. Careful tuning of hyperparameters of each baseline model is needed to ensure a fair comparison.
8. The "Details on ADNI dataset preprocessing" paragraph in Appendix A.1.1 duplicates the reprocessing steps for the MIMIC dataset. In addition, it should be explicitly clarified in A.1.2 that the MIMIC dataset has a linkage with the Massachusetts State Registry of Vital Records and Statistics to allow analysis regarding out-of-hospital mortality up to one year after hospital discharge.

**Questions:**

Please see my comments above.

---

> ### Author Response · Authors · 2024-11-28
> **Response to reviewer L5fQ (1/4)**
>
> We sincerely thank Reviewer **L5fQ** for acknowledging our work as "significant and worth investigating" and appreciating that "considering both contexts is interesting and important." To address the identified weaknesses, we provide detailed point-by-point responses below.
>
> ---
>
> **[W1: Novelty Compared to FuseMoE]**
>
> We respectfully disagree with this concern. While both studies employ Sparse Mixture-of-Experts (SMoE), the fundamental differences lie in how SMoE is utilized in each work. These distinctions can be summarized from the following two perspectives:
>
> 1. **Goal of the Study and Usage of SMoE**
>     Specifically, FuseMoE employs SMoE in the fusion layer, aiming to effectively fuse multimodal information into the same latent space. In contrast, MoE-Retriever utilizes SMoE prior to the fusion layer, with the primary goal of imputing missing modalities while leveraging both intra-modal and inter-modal contexts. This approach enables the effective imputation of missing modality information. Therefore, we respectfully claim that MoE-Retriever is *not* simply an added value on top of FuseMoE, as it does not rely on any backbone design of FuseMoE. The utilization and purpose of SMoE in MoE-Retriever are fundamentally distinct, focusing specifically on **imputation**.
>
> 2. **Router Design**
>     We clarify that router design is not the main focus of our study, unlike prior works that primarily investigate the router design of SMoE [1, 2, 3, 4]. Instead, our study focuses on how to effectively utilize SMoE for **imputation** in the context of missing modalities. Nonetheless, in Section 4.4 (Table 3) of the main paper, we include an ablation study comparing router designs with different numbers of routers to handle both intra- and inter-modal contexts. The results demonstrate that using multiple routers does not outperform a single-router setup, reinforcing the effectiveness of the current router design in MoE-Retriever.
>
>    In summary, rather than advancing router design, our contribution lies in effectively adopting the SMoE framework for the missing modality scenario. Specifically:
>
>    1. **Input Constitution**: Leveraging both intra- and inter-modal contexts as inputs for SMoE.
>    2. **SMoE Goal**: Enabling the router to automatically identify the most relevant experts for imputing missing modality information based on the provided context and modality-specific experts.
>
>
>  [1] Fan, Dongyang, Bettina Messmer, and Martin Jaggi. "Towards an empirical understanding of MoE design choices." *arXiv preprint arXiv:2402.13089* (2024).
>  [2] Huang, Quzhe, et al. "Harder Tasks Need More Experts: Dynamic Routing in MoE Models." *arXiv preprint arXiv:2403.07652* (2024).
>  [3] Dai, Damai, et al. "Stablemoe: Stable routing strategy for mixture of experts." *arXiv preprint arXiv:2204.08396* (2022).
>  [4] Liu, Tianlin, et al. "Routers in vision mixture of experts: An empirical study." *arXiv preprint arXiv:2401.15969* (2024).
>
>
> ---
>
> **[W2: Clarity of Intra-Modal Context]**
>
> Before diving into Equation (2), we would like to provide a clearer overview. The primary goal is to define a supporting group (G) responsible for selecting intra-modal samples. This supporting group consists of the indices (j) where the modality combination satisfies two conditions:
>
> 1. It possesses the observed modality combination.
> 2. It contains the target (missing) modality, which we aim to impute.
>
> For instance, if a sample has ‘GC’ modalities and we aim to impute the ‘I’ modality (as illustrated in Figure 2), the supporting group should include samples with both ‘GC’ and the missing modality ‘I.’ Consequently, the supporting group consists of samples where the modality combination is ‘IGC’ (e.g., $P_2$) or ‘IGCB’ (e.g., $P_9$).

---

> > ### Author Response · Authors · 2024-11-28
> > **Response to reviewer L5fQ (2/4)**
> >
> > Now the reformulated Equation (2) is as follows:
> >
> > - `Formula`
> >
> >    $$G(j \mid \mathcal{T}, mc) = \\{j \in \\{1, 2, \dots, N\\} \mid mc_{j} \in \mathcal{X}(\mathcal{S} \mid \mathcal{T}, mc) \\}$$
> >    $$\text{where} \\; \mathcal{X}(\mathcal{S} \mid \mathcal{T}, mc) = \\{ S \subseteq \mathcal{M} \mid (mc \subseteq S) \land (\mathcal{T} \in S) \\} \\; ∀T∈M,   ∀mc∈MC$$
> >
> > - `Component`
> >
> >    - **$G(j \mid \mathcal{T}, mc)$:** The set of indices (j) of the supporting group where the target modality ($\mathcal{T}$) and modality combination (mcmc) are given. This supporting group is sampled to select intra-modal examples.
> >    - **$mc_{j} \in \mathcal{X}(\mathcal{S} \mid \mathcal{T}, mc) \\}$:** The modality combination of sample j, which must belong to the set of modality combinations $\mathcal{X}$.
> >    - **$\mathcal{X}(\mathcal{S} \mid \mathcal{T}, mc)$:** The set of modality combinations ($\mathcal{S}$) where the target modality ($\mathcal{T}$) and modality combination (mcmc) are given.
> >    - **$(mc \subseteq S) \land (\mathcal{T} \in S)$:** The modality combination \( mc must be a subset of S, and the target modality ($\mathcal{T}$) must also be a subset of S. The operator $\land$ denotes the logical "and" operation. This constraint ensures that the set of modality combinations $\mathcal{X}$ meets the required conditions.
> >
> > To clarify further, we have highlighted key components in orange in the refined version.
> >
> > ---
> >
> > **[W3: Inter-Modal Context]**
> >
> > We would like to re-emphasize the key motivation behind incorporating inter-modal context: to provide personalized context while supplementing the missing modality. Here, "personalized context" refers to the unique characteristics of each sample (e.g., a patient), as each individual possesses a distinct combination of observed modalities and missing modalities. Specifically, the method integrates the embeddings of observed modalities into the input of the SMoE. For instance, as shown in Figure 2, if PiP_i only contains genetic (G) and clinical (C) modalities, we leverage those embeddings as input.
> >
> > To clarify further, as mentioned in the main paper (lines 256–257), we hypothesized that patients with genetic and clinical modalities but lacking imaging modalities could be indicative of the early stages of Alzheimer’s disease (AD). This is because less invasive and more accessible modalities (e.g., clinical and genetic) are often prioritized in early-stage diagnoses. To strengthen this argument with real-world evidence, we refer to the following studies:
> >
> > 1. **Figure 2 in [1]** highlights the sequential diagnostic process:
> >    - Step 1 (Detect): Relies on patient history, medical history, and disease history (clinical modality).
> >    - Step 2 (Assess/Differentiate): Employs genotyping and blood tests (genetic modality).
> >    - Step 3 (Diagnose): Incorporates structural imaging like MRI and PET.
> >       This process underscores the prioritization of clinical and genetic modalities in earlier stages.
> > 2. **[2]** discusses the limitations of imaging in early diagnosis due to its high cost and lack of specificity in identifying early pathological changes. Many early-stage cases are asymptomatic, further emphasizing the reliance on non-imaging modalities for early diagnosis.
> >
> > These observed modality-specific characteristics are uniquely captured through the inclusion of inter-modal context, complementing the intra-modal context, which primarily borrows information from other samples to address the missing modality.
> >
> > To verify the importance of inter-modal context, we refer to the results in Table 3 of Section 4.4 (Ablation Study), where the performance of MoE-Retriever is compared with and without intra- and inter-modal contexts:
> >
> > | Model Variants          | Accuracy (%)     | F1 Score (%)     |
> > | ----------------------- | ---------------- | ---------------- |
> > | MoE-Retriever           | **64.52 ± 2.55** | **63.80 ± 2.96** |
> > | w/o Intra-Modal Context | 61.26 ± 2.33     | 61.80 ± 1.67     |
> > | w/o Inter-Modal Context | 60.97 ± 1.50     | 61.60 ± 0.78     |
> >
> > The results demonstrate a significant drop in performance when inter-modal context is excluded, even more than when intra-modal context is omitted. This underscores the critical role of inter-modal context in the design of MoE-Retriever. Importantly, the optimal performance is achieved when both intra-modal and inter-modal contexts are incorporated.
> >
> >  [1] Porsteinsson, Anton P., et al. "Diagnosis of early Alzheimer’s disease: clinical practice in 2021." *The Journal of Prevention of Alzheimer's Disease* 8 (2021): 371–386.
> >  [2] van Oostveen, Wieke M., and Elizabeth CM de Lange. "Imaging techniques in Alzheimer’s disease: a review of applications in early diagnosis and longitudinal monitoring." *International Journal of Molecular Sciences* 22.4 (2021): 2110.

---

> > > ### Author Response · Authors · 2024-11-28
> > > **Response to reviewer L5fQ (3/4)**
> > >
> > > ---
> > >
> > > **[W4: P' in Eq. (3)]**
> > >
> > > In Equation (3), $\mathbf{P'}{i\text{intra}}$ represents the intra-modal embedding, and $\mathbf{P'}{i\text{inter}}$ represents the inter-modal embedding for sample i. As noted in line 295 of the main paper, $\mathbf{P'}$ is obtained by passing the modality-specific embedding ($\mathbf{P}$) through a Multi-Head Attention (MHA) mechanism. Specifically, as mentioned in line 296, $\mathbf{P'}=MHA(\mathbf{P})$.
> > >
> > > The union operation indicates that for the input to the SMoE, we include both intra-modal and inter-modal embeddings. This design aligns with our primary motivation of integrating both contexts to effectively retrieve the missing modality. To ensure clarity for all readers, we have provided an explicit explanation of the union operation in the refined version.
> > >
> > > ---
> > >
> > >
> > >
> > > **[W5. Parameterization of each expert]**
> > >
> > > For the parameterization of each expert, we followed common practices in the SMoE domain [1, 2, 3, 4]. Specifically, we employed a one-layer Feed-Forward Network (FFN) with random initialization for each expert. To enhance clarity, we have highlighted this expert initialization in the refined version using orange text.
> > >
> > > [1] Shazeer, Noam, et al. "Outrageously large neural networks: The sparsely-gated mixture-of-experts layer." *arXiv preprint arXiv:1701.06538* (2017).
> > > [2] Riquelme, Carlos, et al. "Scaling vision with sparse mixture of experts." *Advances in Neural Information
> > > Processing Systems* 34 (2021): 8583-8595.
> > > [3] Chen, Tianlong, et al. "Sparse moe as the new dropout: Scaling dense and self-slimmable transformers." *arXiv preprint arXiv:2303.01610* (2023).
> > > [4] Lin, Bin, et al. "Moe-llava: Mixture of experts for large vision-language models." *arXiv preprint arXiv:2401.15947* (2024).
> > >
> > > ---
> > >
> > >
> > >
> > > **[W6. Meaning of Input Token]**
> > >
> > > The term "input token" refers to the input of the Sparse Mixture-of-Experts (SMoE), which corresponds to the embedding of each modality in each sample. The rationale behind using "input token" and "embedding" interchangeably in the paper is based on convention in the SMoE literature within vision and language domains [1, 2, 3], where these terms are commonly used synonymously.
> > >
> > > To clarify further with an example: if a sample has two observed modalities, the modality-specific encoder would produce two embeddings, with each embedding corresponding to a token.
> > >
> > > [1] Shazeer, Noam, et al. "Outrageously large neural networks: The sparsely-gated mixture-of-experts layer." *arXiv preprint arXiv:1701.06538* (2017).
> > > [2] Chen, Tianlong, et al. "Sparse moe as the new dropout: Scaling dense and self-slimmable transformers." *arXiv preprint arXiv:2303.01610* (2023).
> > > [3] Lin, Bin, et al. "Moe-llava: Mixture of experts for large vision-language models." *arXiv preprint arXiv:2401.15947* (2024).

---

> > > > ### Author Response · Authors · 2024-11-28
> > > > **Response to reviewer L5fQ (4/4)**
> > > >
> > > > ---
> > > > **[W7: Careful Tuning of Hyperparameters]**
> > > >
> > > > We acknowledge the reviewer’s concern that the optimal hyperparameters from the original paper may not necessarily yield the best performance across different datasets. However, we would like to emphasize that it is a common practice to use the original paper’s best-reported hyperparameters for baseline comparisons, as demonstrated in works like FuseMoE [1] and ShaSpec [2].
> > > >
> > > > Nonetheless, within the rebuttal period, we conducted additional experiments to verify whether careful hyperparameter tuning could lead to improved results. Specifically, we tested combinations of learning rate and hidden dimensions for the best-performing baseline, ShaSpec, on the CMU-MOSI dataset (with V+A+T modalities). The performance, evaluated using a single seed, is summarized below:
> > > >
> > > > | **Learning Rate** | **Hidden Dimension** |  **Acc.**  |  **F1**  |  **AUROC** |  **PRAUC** |
> > > > |:-----------------:|:--------------------:|:-------------:|:------------:|:------------:|:------------:|
> > > > |    1e-4    |     64     | 64.58 ± 1.88 | 64.25 ± 1.55 | 70.51 ± 0.43 | 63.86 ± 1.18 |
> > > > |    1e-4    |     128     | 65.17 ± 0.80 | 64.87 ± 0.78 | 72.00 ± 2.48 | 64.72 ± 4.57 |
> > > > |    1e-4    |     256     | 63.67 ± 0.76 | 63.61 ± 0.73 | 72.77 ± 0.96 | 66.17 ± 2.24 |
> > > > |    1e-3    |     64     | 63.92 ± 1.37 | 63.66 ± 1.45 | 71.98 ± 2.33 | 65.93 ± 3.33 |
> > > > |    1e-3    |     128     | 64.09 ± 0.07 | 63.59 ± 0.14 | 72.46 ± 1.21 | 67.61 ± 2.29 |
> > > > |    1e-3    |     256     | 64.32 ± 0.61 | 64.18 ± 0.54 | 73.33 ± 1.24 | 66.59 ± 3.51 |
> > > > |    1e-2    |     64     | 64.51 ± 2.61 | 64.33 ± 2.49 | 72.67 ± 2.10 | 65.29 ± 1.75 |
> > > > |    1e-2    |     128     | 65.06 ± 3.25 | 64.63 ± 2.99 | 73.18 ± 2.71 | 66.62 ± 2.82 |
> > > > |    1e-2    |     256     | 63.62 ± 0.50 | 63.29 ± 0.54 | 71.81 ± 0.73 | 63.85 ± 1.50 |
> > > >
> > > > From the results in the table, we did not observe a significant performance boost when comparing our chosen hyperparameters (learning rate = 1e-3, hidden dimension = 128, as mentioned in the paper) with other carefully tuned configurations. This suggests that the hyperparameters used in the original paper serve as a reasonable baseline.
> > > >
> > > > Furthermore, the single-run performance of ShaSpec, even with careful tuning, does not surpass that of MoE-Retriever (65.21 ± 2.72), indicating that the upper bound achievable through hyperparameter tuning still falls short of the performance of our proposed method.
> > > >
> > > > We have included this experiment in the refined version in Appendix A.3. For the final version, we will ensure the inclusion of the complete hyperparameter tuning results.
> > > >
> > > > ---
> > > >
> > > > **[W8. Details on ADNI dataset preprocessing]**
> > > >
> > > > We thank the reviewer for highlighting the duplication in the description of the ADNI dataset preprocessing. We followed the same preprocessing procedure for the ADNI dataset as described in Flex-MoE [1], and in the refined version, we have explicitly clarified the usage of this preprocessing.
> > > >
> > > > Additionally, as suggested by the reviewer, we have explicitly detailed the data preprocessing steps for the MIMIC dataset in Appendix 1.2 of the refined version.
> > > >
> > > > 1. Yun, Sukwon, et al. "Flex-MoE: Modeling Arbitrary Modality Combination via the Flexible Mixture-of-Experts." *arXiv preprint arXiv:2410.08245* (2024).
> > > >
> > > > ---
> > > >
> > > > We sincerely appreciate the reviewer **L5fQ** time and effort in reviewing our paper. If you have any remaining concerns, please feel free to reach out. We would be more than happy to address them and are fully prepared to provide our best responses until the deadline ends.
> > > >
> > > > Best,
> > > > Authors

---

> ### Comment · Reviewer_L5fQ · 2024-11-29
> **Thank you for the response**
>
> I appreciate the responses from the authors. Most of my concerns have been alleviated. Particularly, with the updated notations, the equations are now readable and understandable. However, while I agree that the proposed method focuses on addressing the missing modality problem, I think the technical contribution is still a bit incremental. I tend to keep my rating unchanged.

---

> ### Author Response · Authors · 2024-12-01
> **Follow up Response to reviewer L5fQ**
>
> We are glad that reviewer **L5fQ**’s primary concerns, such as updated notations and equations, have been alleviated. Taking this opportunity, we would like to further clarify the technical contributions of our work, which may not have been fully conveyed previously.
>
> ---
>
> **[Follow-up Question: Technical Contribution in This Study]**
>
> We respectfully argue that **MoE-Retriever’s** contribution is not incremental due to its unique characteristics, which we outline as follows:
>
> 1. `Why MoE-Retriever Matters`:
>    - Compared to existing works that either do not consider or solely rely on intra-modal or inter-modal contexts during feature retrieval, MoE-Retriever **incorporates both contexts simultaneously**. Furthermore, while existing approaches typically rely on either a single learnable embedding or all embeddings for retrieval, our method focuses on leveraging only the **relevant embeddings**, where the Sparse Mixture-of-Experts (SMoE) design becomes essential.
>
> 2. `How MoE-Retriever Works`:
>    - MoE-Retriever introduces a novel framework for effective imputation comprising the following three components:
>      - **Intra-Modal Context**: Defines a supporting group that includes samples with observed modalities and a missing modality, serving as a sufficient condition to capture intra-modal similarities across samples.
>      - **Inter-Modal Context**: Utilizes observed modalities from each sample to provide personalized semantic information unique to each sample.
>      - **Context-Aware Routing Policy**: Leverages intra- and inter-modal inputs to dynamically determine the top-k missing modality-specific experts responsible for retrieving the most relevant embeddings for the missing modality.
>
> 3. `What the Advanced MoE-Retriever Design Brings`:
>      Through fine-grained design exploration, we found that:
>      - Using similarities as probabilities for multinomial sampling (rather than random sampling within the supporting group) **did not improve performance**, as detailed in **[W2: Architecture - Incorporating Similarity Measure]** [(link)](https://openreview.net/forum?id=j9DbobO0mY&noteId=4xd3jl16ET).
>      - Increasing complexity by introducing intra-modal-specific routers or increasing the number of experts and routers also **failed to deliver significant performance gains**, as discussed in **[W3: Inter-Modal Context]** [(link)](https://openreview.net/forum?id=j9DbobO0mY&noteId=mWnQC0EizD) and in Table 3 of the main paper.
>      - Replacing the fusion layer with an SMoE layer did not enhance performance.
>
>     These results suggest that the **current simple design space of MoE-Retriever is a “sweet spot”**—balancing simplicity and effectiveness.
>
> In summary, MoE-Retriever tackles imputation through the lens of SMoE by:
> - Modeling both intra- and inter-modal contexts with a context-aware routing policy,
> - Utilizing a simple yet effective design that outperforms more complex alternatives,
> - Making meaningful contributions to the multi-modal learning field with its unique focus on imputation.
>
> ---
>
> We hope this explanation further clarifies the technical contributions of our work, helping to positively influence reviewer **L5fQ**’s final evaluation.
>
> Best,
> Authors

---

> ### Author Response · Authors · 2024-12-03
> **Final Reminder for Reviewer L5fQ's Consideration**
>
> Dear reviewer **L5fQ**,
>
> We sincerely thank you once again for your insightful engagement and the constructive feedback you have shared during the review process. Your thorough comments and thoughtful follow-up questions have played a key role in enhancing the clarity and overall quality of our paper. We deeply value the time and effort you have devoted to carefully assessing our submission.
>
> As the review period enters its final hours, we humbly ask if you might kindly consider revisiting your score once more. We deeply value your feedback regarding the technical contributions of our work and have made a concerted effort to address these concerns in our follow-up responses. In particular, as a `pioneering use of SMoE for missing modality imputation`, we clarified how MoE-Retriever uniquely leverages intra- and inter-modal contexts through a context-aware routing policy, distinguishing it from prior fusion-layer-focused works like FuseMoE.
>
> Borrowing this last moment, given the balanced position of MoE-Retriever among the ratings, we **kindly ask if you might consider offering your final support—such as a slight increase—which could meaningfully impact the outcome of this paper**. If you find our responses and above revisions satisfactory, even a modest increase in your score would mean a great deal to us at this crucial situation. However, we fully understand and respect your judgment and sincerely appreciate the thoughtful consideration you have already provided.
>
> All in all, we sincerely thank you again for your insights and constructive feedback, which have significantly enhanced our work during this rebuttal period.
>
> Best,
> Authors

---

### Author Response · Authors · 2024-11-28
**General Response**

Dear Reviewers,

We extend our sincere gratitude for your thorough review and valuable feedback on our paper. We are truly encouraged by your recognition of the positive aspects of our work, including `an important and well-motivated problem` (all four reviewers), `interesting and reasonable architecture design` (Reviewers **L5fQ**, **3etg**, and **UWni**), `extensive empirical evaluation` (Reviewers **L5fQ**, **UWni**, and **nRr5**), and `a clearly written paper with available code` (Reviewer nRr5).

In addition to addressing your thoughtful comments point-by-point on the OpenReview forum, we have made the following updates in the newly uploaded version of the paper (revisions are highlighted in orange):

1. **Clear Presentation of Intra-Modal Context** (Reviewer **L5fQ**): Clarifications and examples have been added to `Section 3.2.1`.

2. **Missing Citations to Previous Work** (Reviewer **3etg**): Citations for FuseMoE experiment results (`Section 4.2`), the ADNI missingness figure (`Section 4.3`), and the data preprocessing procedure (`Appendix A.1`) have been included.

3. **Ablation Study for Different Dropping Ratios** (Reviewer **3etg**): Additional experiments were conducted to assess performance robustness under different dropping ratios, with results included in `Appendix A.3`.

4. **Additional Metrics: AUROC and PRAUC** (Reviewers **3etg**, **nRr5**): Results for AUROC and PRAUC on MIMIC datasets have been added in `Appendix A.4`.

5. **Careful Hyperparameter Tuning of Baselines** (Reviewer **L5fQ**): Additional experiments were conducted to tune the ShaSpec baseline model at `Appendix A.5`.

6. **Analysis of Performance Drop on CMU-MOSI Dataset** (Reviewer **3etg**): We analyzed why models using three modalities (V+A+T) perform worse than those using two modalities (A+T), identifying gradient conflicts as the root cause at `Appendix A.6`.

7. **Effectiveness of Retrieval in MIMIC Dataset** (Reviewer **nRr5**): t-SNE plot of patient embeddings before and after retrieval is demonstrated in `Appendix A.7`.


For quick access to the updated tables and figures in the Appendix, please refer to the link below:
- **Link: https://anonymous.4open.science/r/moe-retriever-rebuttal-24A8/README.md**

We have made diligent efforts to address all concerns raised and are committed to further engaging with any additional inquiries you may have.

Best,
Authors

---

### Meta-Review · Area_Chair_mLP2 · 2024-12-21

**Metareview:**

The authors propose a new method to address missing modalities, called MoE-Retriever, using attention and SparseMoE to retrieve embedding candidates.

While the reviewers found the method to be well motivated and generally sound, issues were raised concerning its novelty (as it is a combination of existing techniques).

Some of the reviewer questions concerning how the method operates were successfully addressed by the authors, however, the key point of it being incremental and simply a use of the existing MoE methods was still seen as an issue by two of the four reviewers.

Overall, based on the reviews and the rebuttal, I have to agree with them. Until more contributions are brought forward, my assessment is that this paper will fall short of the bar for acceptance in top tier conferences.

**Additional Comments On Reviewer Discussion:**

No additional comments. The problem with this paper is lack of novelty, which no amount of additional experiments or explanations can fix. All 4 reviewers responded to author comments.

---

### Decision · Program_Chairs · 2025-01-22

Reject